# Electrochemical and Photoelectrochemical Immunosensors for the Detection of Ovarian Cancer Biomarkers

**DOI:** 10.3390/s23084106

**Published:** 2023-04-19

**Authors:** Ezinne U. Ekwujuru, Abimbola M. Olatunde, Michael J. Klink, Cornelius C. Ssemakalu, Muntuwenkosi M. Chili, Moses G. Peleyeju

**Affiliations:** 1Department of Biotechnology and Chemistry, Vaal University of Technology, Vanderbijlpark 1911, South Africa; 2Department of Chemistry, University of Ibadan, Ibadan 200284, Nigeria; 3Centre for Academic Development, Vaal University of Technology, Vanderbijlpark 1911, South Africa

**Keywords:** photoelectrochemical immunosensors, cancer biomarkers, ovarian cancer, electrochemical sensing

## Abstract

Photoelectrochemical (PEC) sensing is an emerging technological innovation for monitoring small substances/molecules in biological or non–biological systems. In particular, there has been a surge of interest in developing PEC devices for determining molecules of clinical significance. This is especially the case for molecules that are markers for serious and deadly medical conditions. The increased interest in PEC sensors to monitor such biomarkers can be attributed to the many apparent advantages of the PEC system, including an enhanced measurable signal, high potential for miniaturization, rapid testing, and low cost, amongst others. The growing number of published research reports on the subject calls for a comprehensive review of the various findings. This article is a review of studies on electrochemical (EC) and PEC sensors for ovarian cancer biomarkers in the last seven years (2016–2022). EC sensors were included because PEC is an improved EC; and a comparison of both systems has, expectedly, been carried out in many studies. Specific attention was given to the different markers of ovarian cancer and the EC/PEC sensing platforms developed for their detection/quantification. Relevant articles were sourced from the following databases: Scopus, PubMed Central, Web of Science, Science Direct, Academic Search Complete, EBSCO, CORE, Directory of open Access Journals (DOAJ), Public Library of Science (PLOS), BioMed Central (BMC), Semantic Scholar, Research Gate, SciELO, Wiley Online Library, Elsevier and SpringerLink.

## 1. Introduction

Cancerous growth in any part of human body is one of the leading causes of death in recent years. Among gynaecologic cancers, ovarian cancer has been indicated to be the deadliest [1]. Ovarian cancer (OC) is a group of diseases that emerges from the ovaries or associated parts of the fallopian tubes and peritoneum [2]. About 250,000 cases of ovarian cancer (OC) are diagnosed annually with 140,000 casualties worldwide [3]. In America, 21,750 cases were diagnosed in 2020 with about 13,940 deaths [4], while about 18,000 cases were reported in Sub–Saharan Africa with around 13,000 mortalities [5]. The incidence rate of OC increases with age and post–menopausal status [6]. In women below 50 years of age, the incidence is about 4.7 per 100,000 while in older women of 50–64 years, the incidence is about 29.6 per 100,000 [6]. Postmenopausal women are more prone to OC due to the strong relationship between age advancement and disease incidence [4]. Compared to breast cancer, which is more prevalent, OC is up to three times more fatal than breast cancer and the mortality rate is predicted to increase remarkably by the year 2040 [7]. This high rate of mortality is attributed to the asymptomatic and secret growth of the tumor, late appearance of symptoms, and deficient proper screening which leads to late diagnosis [7,8].

Early diagnosis is paramount to successful cancer treatment and cure. According to the International Federation of Obstetrics and Gynecology (FIGO), the survival rate for OC patients at stage 1 of the disease is high. Up to 90% of patients can be successfully cured at this early stage; whereas less than 20% can be cured once the disease has reached advanced stages [9]. Although no screening method has been recommended yet for early detection of OC [10], biomarkers—biological molecules that are elevated or produced during a pathological process—help in predicting the early occurrence or recurrence of cancer as well as treatment progress. Enzyme–linked immunosorbent assay (ELISA), liquid chromatography–mass spectroscopy (LC–MS), radioimmunoassay (RIA), surface plasmon resonance (SPR) immunoassay, electrochemiluminescence (ECL), fluorescence in–situ hybridization and fluorescence spectroscopy are some of the conventional methods that are in use for the early detection of biomarkers [11,12]. These techniques have excellent accuracy, but are costly, time–consuming, unsuitable for point–of–care testing, and sophisticated, thus require trained personnel [12,13,14,15]. In addition, some of these techniques are unreliable in detecting ultra–low levels of biomarkers when the cancer is in the early stages, hence the need for highly reliable, non–invasive, low cost, point–of–care diagnostic tools [12] that are also capable of being specific and sensitive [4] with ultralow detection levels [11].

Electrochemical (EC) biosensors are used as non–invasive point–of–care tools. They generally have three major components; a biorecognition element which detects the target biomolecule in a sample, a transducer which converts the biological reactions to a measurable electrical output, and a signal processor which processes the data and displays results [16]. Electrochemical immunosensors use the electrical signals produced from the highly specific biorecognition reaction that exists between an antigen and an antibody which is the capture material to quantitatively measure the target molecule [17]. Conventionally, EC sensors uses a 3–electrode configuration system which can be flexible and portable with a small amount of electrolyte [18] and sample requirements. The materials used for its construction are usually cheap, including the electrodes which are also configured using simple electronics [19]. These basic characteristics, and others, have conferred some advantages to EC biosensors. These include multi–analyte testing capability, ease of use, portability, and low cost while maintaining accuracy and reliability [16,20]. In electrochemical biosensing, high electrical conductivity, excellent biocompatibility and active surface area of the materials used are important factors to consider [18], hence the importance of making this platform sensitive, precise [21], with a very high specificity and rapid response time [16,22]. However, the sensitivity of EC biosensors can be affected by the background noise caused by side reactions. In order to minimize the effect of the background noise and enhance the sensitivity of EC biosensors, light energy has been incorporated into the detection system to produce a photoelectrochemical sensing platform [23].

Photoelectrochemical (PEC) biosensing combines photo irradiation with electrochemical detection, hence have the advantages of both optical and electrochemical methods [24,25]. The separation of the excitation source and detection signals reduces the undesirable background noise, thereby increasing its sensitivity [25] and making lower limits of detection for analytes achievable [26]. It uses semiconductors which utilizes light energy to generate electron–hole pairs while also facilitating the rate of electron transfer which are utilized in PEC sensing [27]. PEC biosensing is a recently developed technique with promising applications in the field of bioanalysis [28]. Because it is still in its infancy, fewer PEC immunosensors have been developed over the years especially for the detection of ovarian cancer biomarkers. This review discussed the most relevant electrochemical and photoelectrochemical immunosensors developed in the very recent past (2016–2022) for the detection of the major ovarian cancer biomarkers.

## 2. EC and PEC Immunosensors for the Determination of Carcinoembryonic Antigen (CEA)

### 2.1. EC Immunosensors for OC Biomarkers Detection

The detection principle of this type of immunosensor is based on the signals produced during the biorecognition events between the target analyte (antigen) and its bioreceptor (antibodies) through an electrical interface(s). These electrical signals are processed and recorded in the electrochemical reader. Figure 1 shows the basic components of a typical EC immunosensor. EC immunosensors can either be label–free or sandwich–type.

#### 2.1.1. Label–Free EC Immunosensors for OC Biomarkers Detection

This type of biosensor does not use labels but can directly detect the target analyte through the binding of the target analyte with the biorecognition element [29]. This technique is considered to be more efficient due to absence of the problems that has been identified with the use of labels, such as time consumption and modification of the binding sites of biomolecules which may interfere with sensitivity [30].

##### Label–Free EC Immunosensors for Carcinoembryonic Antigen (CEA) Detection 

CEA is a glycoprotein from the immunoglobulin family [31]. It is normally produced in the gastrointestinal tissue during prenatal development but it stops being produced before birth [32]. However, low levels are still found in healthy human serum [32] with an estimated level of 2.5 ng/mL [33]. CEA is an important oncomarker for ovarian and breast cancers [16,33,34,35] and many other cancer types such as gastric, lung, pancreatic, colorectal cancers [36] and cervical carcinoma [37]. It can also be used as a marker to directly evaluate curative effects, recrudescence, and metastasis [37]. Currently, it is widely used as a validated prognostic blood–based protein biomarker [38]. Due to its universality, it is the most studied cancer biomarker, hence the reason for the overwhelming number of immunosensors that have been developed so far for its detection. For example, Gao et al. [39] designed a label–free, highly stable, sensitive and selective EC immunosensor for CEA detection. The group used nanocomposites of gold nanoparticles (Au NPs) and Nile blue A (NB) hybridized electrochemically reduced graphene oxide (NB–ERGO) to functionalize a glassy carbon electrode. Then, CEA antibody was immobilized on the modified electrode to form the immunosensor antiCEA/AuNPs/NB–ERGO, which achieved a linear range of 0.001 to 40 ng/mL and detection limit of 0.00045 ng mL^−1^. Satisfactory results were obtained when the developed immunosensor was used for the quantification of CEA in clinical serum samples. The good electrochemical performance of the biosensor was due to the good electrons–transfer ability of AuNPs/NB–ERGO. In addition, the large amount of Au NPs loading on the surface of NB–ERGO permitted more anti–CEA to be absorbed on the electrode surface, which in turn increased access to the antigen greatly enhanced the electrochemical performance. Again, Wang et al. [40] prepared a simple selective label–free CEA electrochemical immunosensor based on flower–like Ag/MoS_2_/rGO nanocomposite. The team used a prepared solution of the nanocomposite to modify a GCE, after which anti–CEA was dropped onto the modified electrode, which was conjugated to the surface of Ag NPs through amino groups. The nanocomposite acted as a signal amplifier where molybdenum disulfide (MoS_2_), reduced graphene oxide (rGO) and Ag NPs synergistically enhanced the sensitivity of the immunosensor through the catalytic reduction of H_2_O_2_. The anti–CEA/Ag/MoS_2_/rGO biosensor achieved a wide detection range of 0.01 pg/mL to 100 ng/mL and low detection limit of 1.6 fg/mL using amperometric i–t curve detection technique. It also exhibited acceptable reproducibility, selectivity, and stability and have good agreement with the ELISA method. The sensor could also be applied successfully in the detection of CEA in human serums.

For a more rapid detection of CEA, Lin and co–workers [41] constructed a special disposable label–free immunosensor which exhibited a limit of detection of 4.25 pg/mL and a limit of quantification of 12.89 pg/mL with a concentration linear range from 0.01 to 10 ng/mL. The fabrication was based on modifying screen–printed carbon electrode (PE) with graphene–zirconia (GZ) nanocomposite functionalized with 1–Pyrenebutyric–Acid–N–hydroxysuccinimide–ester (PYSE) which served as a crosslinker for the immobilization of anti–CEA (Ab). Skim milk (SM) was employed as an active–site blocker. The developed SM/Ab/Gz–PYSE/PE biosensor was applied to human serum sample with good results. Electrochemical measurements were achieved using cyclic voltammetry and electrochemical impedance spectroscopy (EIS) in the presence of [Fe(CN)_6_]^3–/4–^ redox species. In the same vein, Rizwan and co–authors [42] modified a GCE using a nanocomposite of Au NPs, carbon nano–onions (CNOs), single–walled carbon nanotubes (SWCNTs) and chitosan (CS) prepared using the one–pot preparation method, followed by the addition of anti–CEA into the modified electrode (Figure 2). The developed label–free electrochemical immunosensor–AuNPs/CNOs/SWCTNs/CS/anti–CEA was used for the quantitative detection of CEA. The fabrication process was monitored using CV and SWV techniques, employing [Fe(CN)_6_]^3–/4–^ as a mediator solution. A low detection limit of 100 fg mL^–1^, and linear detection range from 100 fg mL^–1^ to 400 ng mL^–1^, were reached. The as–prepared immunosensor was highly selective, specific, and interference–resistant with good stability and significant potential in detecting CEA in real serum samples.

Interestingly, in 2016 Wang and his group [37] developed a highly sensitive and low–cost microfluidic paper–based label–free electrochemical immunosensor for detecting CEA. A fabricated screen–printed working electrode (SPWE) containing the microfluidic paper was modified with a nanocomposite of amino– functionalized graphene (NH_2_–G), thionine and gold nanoparticles (NH_2_–G/Thi/AuNPs). Thionine, which acted as the electrically active substance by generating electric current through redox reactions on the surface of the electrode, was attached to the surface of NH_2_–G through noncovalent π–π stacking interactions. NH_2_G amplified the signal through enhancement of electron transfer due to its excellent conductivity and good biocompatibility. The immobilization of AuNPs, which aided fast electron transportation, was made possible by its interactions with amino groups. The nanocomposite was used to immobilize anti–CEA and to enhance detection sensitivity. Detection was based on the reduced current response of thionine due to the antibody–antigen immunocomplex formation, which was proportional to the concentrations of antigens. A linear response range of 50 pg mL^–1^ to 500 ng mL^–1^ and a limit of detection of 10 pg mL^–1^ was realized with a corresponding correlation coefficient of 0.996. The developed immunosensor showed good repeatability and excellent selectivity with potentials in clinical usage. Wang and his group [43] reported an EC immunosensor developed by using bismuth oxide doped with molybdenum using a one–pot synthesis method and then modified with NH_2_ group and AuNPs to form Au@Bi_2_MoO_6_ nanotubes with a tremella–like crystal structure (Figure 3). This was used as the electroactive material to coat the glassy carbon electrode via the drop–drying method. Subsequently, anti–CEA was immobilized using Gallic acid as a linker, washing unbound antibody with PBS and blocking inactive sites using BSA to complete the fabrication of the immunosensor. The immunosensor was used to quantify CEA concentrations, which gave a wide linear range of 1 pg/mL to 1µg/mL and a detection limit of 0.3 pg/mL measured with SWV. The sensitivity of the biosensor was high, with acceptable selectivity and stability, and could be used in real samples for the detection of CEA.

Cao et al. [44] established a label–free ultrasensitive EC immunosensor for the rapid detection of CEA, which exhibited an LOD of 33.11 fg/mL with a linear concentration ranging from 100 fg/mL to 100 ng/mL. The construction was based on a cuprous sulfide (Cu_2_S) and palladium nanoparticles (Pd NPs) nanocomposite with CuO. Pd NPs were formed on Cu_2_S through in situ growth, while CuO was synthesized via partial oxidation of Cu_2_S by hydrogen peroxide. The snowflake–like Cu_2_S/Pd/CuO nanocomposites were used to modify a GCE, which showed excellent catalytic performance towards hydrogen peroxide reduction that amplified signals. Anti–CEA was then added to the modified electrode for CEA detection which resulted in a highly specific, repeatable, and stable biosensor that also has potentials in clinical applications for detecting CEA antigen. In 2019, Idris et al. [45] constructed a simple EC immunosensor based on a polypropylene imine dendrimer (PPI) and carbon nanodots (CNDTs) nanocomposite decorated on an exfoliated graphite electrode (EG) for CEA detection. The CNDTs, which were synthesized from oats through pyrolysis, was drop–coated onto a carbon nanotube– modified exfoliated graphite electrode. Then, the PPI, which provided a biocompatible high surface area and functional groups, was electrochemically deposited onto the EG–CNDTs. Anti–CEA was then immobilized using glutaraldehyde as a crosslinker, followed by addition of BSA used for blocking remaining active sites. The produced biosensor had good selectivity for CEA detection with good reproducibility and a linear detection range from 0.005 to 300 ng/mL and LOD of 0.00145 ng/mL, determined using DPV. Another simple method for EC immunosensing of CEA was developed by Paimard et al. [46] through direct electrospinning of core–shell honey nanofibers (HNF) onto the surface of a carbon–paste electrode (CPE) which was made by mixing graphite powder and mineral oil as shown in Figure 4. Then, gold nanoparticles (GNPs) were electrodeposited onto the modified electrode followed by functionalization with multi wall carbon nanotubes (MWCNTs). The modified electrode surface was then treated with EDC and NHS to activate the carboxyl groups on the MWCNTs surface. CEA antibody was then immobilized and used for CEA detection. Using CV and EIS for electrochemical measurements, a wide linear concentration range of 0.4–125 ng/mL and a low detection limit of 0.09 ng/mL was achieved. These results were consistent with those obtained using of ELISA. In addition, the as prepared immunosensor possessed good selectivity, sensitivity, stability, and reproducibility, which was due to the high surface area, greater electrical conductivity, and electron–transfer capability of MWCNTs, GNPs and HNF used in the fabrication process.

Shamsuddin et al. [38] proposed a label–free EC immunosensor based on the use of a novel non–conducting polymer. First, the group electropolymerized octopamine onto the working electrode of a screen–printed gold electrode (SPGE). Then, oxidized antibodies were coated onto the functionalized electrode. A covalent bond was formed between the aldehyde group of the antibodies and amine groups on the polyoctopamine (POct)–modified electrode. The functionalized electrodes (anti–CEA–mAb/POct/SPGE; anti–CEA–pAb/POct/SPGE) were then used to detect the CEA antigen. Using monoclonal antibodies (mAb), the LOD obtained was 10.8 fM while LOD for polyclonal antibodies (pAb) was 9.08 fM. In addition, Singh and colleagues [47] detected CEA using a disposable label–free EC immunosensor, utilizing an ITO electrode decorated with AgNPs–SiO_2_ composites (Ag@SiO_2_ NPs). The functionalized electrode was used to immobilize anti–CEA then HRP and BSA were employed to block the non–specific binding sites. In addition, HRP enzyme elevated detection signals through its catalytic activity. The developed ITO/Ag@SiO_2_ NPs/anti–CEA/HRP immunosensor was used to quantify CEA antigen in the range of 0.5–10 ng/mL and LOD of 0.01 ng/mL. the immunosensor possessed other appealing characteristics such as excellent specificity, selectivity, stability and reproducibility.

The electrocatalytic effect resulting from combining two metals (mostly noble metals) to form bimetallic nanoparticles is higher than the individual metals because of the dual catalytic effect [48,49]. Yang and co–workers [48] utilized this advantage by exploring the biocompatibility, excellent electrocatalytic activity, and high conductivity of platinum nanoparticles and palladium nanoparticles to develop a label–free EC immunosensor for the detection of CEA. The group used nanocomposites of hydrothermally synthesized nitrogen–doped graphene QDs (N–GQDs) supported PtPd bimetallic nanoparticles functionalized with Au nanoparticles (PtPd/N–GQDs@Au) through a self–assembly method to modify the electrode. This platform was used as the transducer to immobilize capture antibodies effectively and also as a signal amplifier. The nanocomposite exhibited excellent electrocatalytic activity towards hydrogen peroxide reduction due to the synergistic effect present. The immunosensor had a wide linear range of 5 fg/mL to 50 ng/mL and low defection limit of 2 fg/mL with a high selectivity and sensitivity for CEA detection and long–term stability. In addition, Yang and team [50] designed a label–free EC immunosensor that detected CEA in the range of 0.001 ng/mL to 80 ng/mL with detection limit of 0.286 pg/mL. Using one–pot preparation method, the group prepared silver nanoparticles (Ag) functionalized graphene oxide sheet (GO) and treated with dopamine (DA) which acted as a reducing agent and self–polymerized in the process to form Ag/rGO@PDA hybrid. This hybrid was then doped with gold nanoparticles for dual amplification. The Au–Ag/rGO@PDA nanocomposite was used to modify the glassy carbon electrode using drop drying technique. Then, anti–CEA was immobilized for antigen detection. The developed immunosensor has good selectivity, high stability and reproducibility, and can be used as a reference for detecting CEA in clinical practice and medical research. Song et al. [51] used rGO/MoS_2_@PANI to construct a highly sensitive, specific and repeatable EC immunosensor for CEA detection. The rGO/MoS_2_ composite was prepared through ultrasonic dispersion and subsequent hydrothermal treatment before being functionalized with polyaniline (PANI). The rGO/MoS_2_@PANI composite was then used to modify a gold electrode followed by sequential immobilization of anti–CEA and BSA. rGO/MoS_2_ composite enhanced electron transfer and also provided large specific surface area to accommodate more PANI. PANI contains rich sites for antibodies immobilization. The prepared immunosensor was used to quantify CEA antigen in a linear concentration range of 0.001–80 ng/mL with LOD of 0.3 pg/mL. 

##### Label–Free EC Immunosensors for Cancer Antigen 19–9 (CA 19–9) Detection 

CA 19–9 is a glycoprotein linked to ovarian cancer [16,52,53], cervical cancer [54] and other general cancers such as pancreatic, colorectal, gastric and liver cancers [52,53]. Its serum concentration is usually below 37 U/mL in a normal healthy person, while an elevated level is an indicator of possible cancer occurrence [52,53]. Figure 5 shows a label–free EC immunosensing technique for CA19–9 detection recently introduced by Kalyani and colleagues [55]. The researchers used a glassy carbon electrode (GCE) and modified it with a solution of chitosan (CS) marked–nanocomposites of multiwalled– carbon nanotube and magnetite nanoparticles (CS–MWCNT–Fe_3_O_4_). Then, CA19–9 antibody was immobilized onto the functionalized electrode through crosslinking with glutaraldehyde with subsequent coating with BSA (bovine serum albumin). Fe_3_O_4_ which enhanced charge transport and improved the stability of chitosan and MWCNT including the high conductivity of MWCNT and good bio–adsorption of chitosan synergistically improved the sensing performance of the sensor. The fabricated immunosensor, anti–Abs/CA19–9/CS–MWCNT–Fe_3_O_4_ showed a sensitivity of 2.55 µA Pg^−1^ cm^−1^ and was able to detect CA19–9 antigen in the range of 1.0 Pg/mL to 100 ng/mL with limit of detection (LOD) of 0.163 Pg/mL measured with square wave voltammetry (SWV) in the presence of redox marker [Fe (CN)^6^] ^3/4^. 

Also, Wang, et al. [56] derived a special label–free EC immunosensing method for CA19–9 detection which exhibited an extremely low LOD of 10 µU/mL with a wide linear concentration range of 0.1 mU/mL to 10 U/mL using electrochemical impedance spectroscopy (EIS). The group used bimetallic cerium and ferric oxide NPs which were evenly embedded within graphitized mesoporous carbon matrix and calcinated at a temperature of 500 °C. The formed CeO_2_/FeO_x_@mC composites were then used to immobilize anti–CA19–9 for the detection of CA199 antigen. Using a GCE, another label–free EC immunosensor for CA19–9 detection was proposed by Su and his colleagues [53] as depicted in Figure 6. The team used Zn–Co–S/graphene nanocomposites prepared through a one–step hydrothermal method to first functionalize the electrode by dropping method. This formed the electroactive sites for electrochemical reactions. Chitosan and glutaraldehyde were further used to modify the electrode with the later acting as a crosslinker to the antibody. The developed anti–CA19–9/GA/CHIT/Zn–Co–S@G/GCE immunosensor had a linear detection range of 6.3 U/mL to 300 U/mL and LOD of 0.82 U/mL. Characterization of the entire process was carried out using EIS and amperometry. In 2017, Huang et al. [52] designed a simple and environmentally friendly, highly sensitive, stable and reproducible label–free EC immunosensor for CA19–9 quantification. They modified a GCE with Au NPs and employed polythionine–Au NPS composites (Au NPs@PThi) as a crosslinking agent which was drop casted on the pretreated GCE surface. CA19–9 antibody was then immobilized on the functionalized electrode followed by BSA to form the biosensor, BSA/anti–CA19–9/Au NPs/Au NPs@PThi/GCE. The linear detection range of the sensor was between 6.5 to 520 U/mL with LOD of 0.26 U/mL which were determined using differential pulse voltammetry (DPV) and EIS.

Ibáñez–Redín and group [57] established a novel simple and cheap disposable EC immunosensor for CA 19–9 detection by utilizing an SPCE functionalized with layer–by–layer films of carbon black (CB) and polyelectrolytes (PEL). This served as a matrix for CA 19–9 antibodies immobilization and subsequent antigen capturing. Electrochemical sensing performance was carried out with DPV which recorded an LOD of 0.07 U/mL and linear concentration range of 0.01 to 40 U/mL. In addition, the result obtained in using the fabricated sensor to determine CA19–9 on cell lysate and human serum samples of cancer patients were in good agreement with that of standard assay. In 2019, Ibáñez–Redín led another group [58] to fabricate an EC immunosensor for the quantification of CA 19–9, this time using a home–made silver screen–printed interdigitated electrodes (SPIDES). The electrode was coated with carbon nano–onions (CNO) followed by modification with graphene oxide (GO) films through adsorption before immobilization of CA 19–9 antibodies through electrostatic interaction between the antibodies and GO. The developed low–cost immunosensor was able to have a low LOD of 0.12 U/mL and linear concentration range of 0.3–100 U/mL which was determined with EIS over an applied potential of 50 mV. In addition, Thapa et al. [59] produced a similar label–free EC immunosensor which had an LOD for CA 19–9 as 0.35 U/mL. The team used a layer–by–layer technique to functionalize interdigitated gold electrodes with polyethleneimine (PEI) and carbon nanotubes (MWCNTs). Then, CA 19–9 antibodies were immobilized onto the modified electrode using EDC/NHS chemistry which activated the carboxylic acid groups on the surface of MWCNTs for CA 19–9 antigen detection. Recently, Wei et al. [60] described another label–free EC immunosensing method for the detection of CA 199 using piranha pretreated gold electrode. The pretreated electrode was coated with a self–assembled layer of 3–mercaptopropionic acid (MPA) and β–mercaptoethanol (ME) hybrid (MPA/ME/Au). Through EDC/NHS activity, anti–CA 199 was immobilized and used for the detection of CA 199 antigen. The fabricated immunosensor exhibited a linear concentration range from 0.05–500 U/mL with LOD of 0.01 U/mL. It also had excellent precision, good accuracy and can be applied in real time detection of CA 199.

##### Label–Free EC Immunosensors for Alpha–Fetoprotein (AFP) Detection 

Alpha–fetoprotein is a glycoprotein normally produced by the liver, yolk sac, and gastrointestinal tract during early fetal life but decreases over time [13,61,62]. It is the major marker for liver cancer [63]. It can also serve as a marker for the detection, diagnosis and monitoring of testicles and ovarian cancers [64]. In a normal adult, the serum level is less than 25 ng/mL [65]. In the quest for a very sensitive transducing matrix for detecting AFP, Liu et al. [66] took full advantage of semiconductor/organic heterointerface. The group developed a sensitive label–free EC immunosensor with good selectivity, reproducibility and stability for the quantification of AFP using aligned gallium nitride (GaN) nanowire arrays synthesized using a simple chemical vapor deposition process. GaN was biofunctionalized with polydopamine (PDA) by dip–coating through self–assembly to enhance charge transfer. Au NPs were used to modify the GaN–PDA hybrids to enable adequate immobilization of the antibodies via covalent bonding. Electrochemical measurements for AFP antigens were carried out using DPV with results of LOD and linear range of detection being 0.003 ng/mL and 0.01 to 100 ng/mL, respectively. In 2017, Wang and his team [67] fabricated another label–free EC immunosensor for detecting AFP based on the use of nanocomposites of graphene oxide–doped ferroferric oxide nanoparticle–decorated gold nanoparticles with toluidine blue tag (TB–Au–Fe_3_O_4_–rGO) to modify a bare GCE. Through the interaction of the biocompatible AuNPs and the amino group of the antibodies, anti–AFP was immobilized and used for the detection of AFP antigen. Toluidine blue was used as the redox probe for electrochemical signaling and ferroferric oxide NPs as an electrocatalyst for the reduction and oxidation of toluidine blue. Graphene oxide is a good adsorbing material with large specific surface area for antibody attachment and also has excellent electrical conductivity that favors electron transfer. These materials synergistically improved the stability and sensitivity of the immunosensor which exhibited an LOD of 2.7 fg/mL with linear concentration range of 1.0 × 10^−5^ ng/mL to 10.0 ng/mL measured with SWV.

##### Label–Free EC Immunosensors for p53 Detection 

The tumor suppressor protein p53 plays a crucial role in the cell cycle by regulating cell activities such as cell growth and proliferation, DNA repair and apoptosis [68,69]. Its loss of function can lead to tumorigenesis and gene mutation as a result of the conformational variations that exist in the p53 protein structure [68,69]. About 50% of all human tumors has p53 mutants [70] and is the most common mutations found in human cancers [71]. The serum level and half–life of p53 protein is increased when there are mutations in the Tp53 gene, degradation of protein or viral cancer genes and as such it can be used for screening, prognosis or monitoring of many cancer types such as lung, head, neck, rectal, oral, prostate and some gynecological cancers such as breast and cervical cancers with high diagnostic performance [72]. It is also used as oncomarker for ovarian cancer [16,71,73]. Hence its early detection is crucial. Hasanzadeh et al. [70] fabricated a label–free EC immunosensor using a gold electrode for the detection of p53. The electrode was first functionalized with poly L–cysteine (P–Cys) through direct electrochemical polymerization of L–cysteine on the electrode which formed covalent bond with the gold surface. Then, graphene quantum dots (GQDs) was electrodeposited onto the modified surface followed by sono–electrodeposition of gold nanoparticles (GNPs). Streptavidin–HRP and BSA were then sequentially coated onto the modified electrode before immobilizing anti–p53 for biorecognition of the antigen. P–Cys was used as the conductive matrix while GQDs/GNPs composite served as dual signal amplifier. In the presence of thionine as electron mediator that facilitates electron transfer, HRP could electrocatalyze the Redox of H_2_O_2_ to amplify current signals. The prepared immunosensor, Au/P–Cys/GQDs/GNPs–streptavidin/HRP/BSA/Ab was able to detect p53 antigen within a linear response range of 5.92 × 10^−4^ to 1.296 pM and lower limit of 6.5 × 10^−2^ fM in unprocessed human plasma using SWV. In 2016, Elshafey et al. [71] also reported a sensitive label–free EC immunosensor for the quantification of anti–p53 in serum based on self–assembled Au NPs onto thiolated electrochemically reduced graphene oxide (ERGO) modified SPCE. The functionalization of the electrode surface with ERGO was aided by first coating the electrode surface with electrografted p–aminophenol organic layer used as a linker. The AuNPs/ERGO interface was effective in immobilizing p53 antigens and enhancing bioactivity and stability of the immunosensor due to its large surface area. Scanning electron microscopy, Raman and X–ray photoelectron spectroscopies were employed for the characterization of the fabrication process while SWV was used to investigate the antigen–antibody reactions. A linear range of 0.1 pg/mL to 10 ng/mL and low detection limit of 0.088 pg/mL was recorded using the developed immunosensor. Again a simple label–free EC immunosensor was proposed by Aydin et al. [69] for the quantitative detection of p53 using an ITO electrode (Figure 7). To modify the electrode surface, the group used chitosan and carbon black composite (Chitosan–CB). p53 antibody was then immobilized through chitosan–glutaraldehyde crosslinking for the sensitive and selective capturing of p53 antigen. The immunosensor exhibited good sensitivity, repeatability and stability with a wide linear concentration range of 0.01 to 2 pg/mL with an LOD of 3 fg/mL which were determined using changes in impedance via EIS and CV.

Also in 2018, Aydin et al. [68] constructed another label–free EC immunosensor for p53 detection using a disposable ITO electrode modified with tetra armed star–shaped poly (glycidyl methacrylate) (Star_PGMA_) with epoxy ends. These epoxy ends covalently bonded to p53 antibody during immobilization to form a reproducible and repeatable immunosensor. Star_PGMA_ was synthesized using atom transfer radical polymerization in the presence of tetra functional initiator. Then, it was used to make a thin film on the electrode via a spin–coating technique. The prepared immunosensor was able to detect p53 antigen in a linear concentration range of 0.02 pg/mL to 4 pg/mL with LOD of 7 fg/mL determined using EIS. It also exhibited excellent selectivity, good reproducibility, acceptable stability, suitable for reusability, and can be successfully used to determine p53 antigen in human serum samples. Ibáñez–Redín and co–authors [74] also developed a simple and disposable label–free EC immunosensor for use in detecting p53. The researchers first coated an SPCE with layer by layer assembled cationic polyethyleneimine (PEI) before modifying it via layer–by–layer assembly with hydrothermally synthesized carboxylated NiFe_2_O_4_ NPs. Through electrostatic interaction with the positively charged PEI layer, the NPs were adsorbed. The modified electrode was then coated with p53 antibodies. The antibodies were adsorbed on the matrix via covalent binding to the carboxylate groups of the carboxylated NiFe_2_O_4_ NPs, activated by using EDC/NHS chemistry. The SPCEs/PEI/NPs–Ab immunosensor was then used for p53 antigen quantification which realized an LOD of 5.0 fg/mL and wide linear concentration range of 1.0 to 10 × 10^3^ pg/mL. The developed immunosensor possessed good selectivity with negligible interference when used in complex samples. Not quite long, Chen et al. [72] fabricated a time–saving label–free EC immunosensor for p53 detection based on a screen–printed gold electrode (SPGE) functionalized with nickel phthalocyanine (NiPc) using drop casting method. Anti–p53 was immobilized onto the surface of the Au/NiPc electrode through the interaction of its imidazole group and Ni^+2^ ion of NiPc. Then, the modified electrode was used to detect p53 antigen in sample following BSA treatment. A linear detection range of 0.1–500 pg/mL was realized using CV for electrochemical measurements in the presence of (K_3_Fe(CN_6_)/K_4_Fe(CN)_6_. The immunosensor had a high sensitivity of 60.65 µA/Log (pg/mL)–cm^2^ with other advantages such as good specificity and selectivity for p53 and fast detection time of 90–150 s as well as being applicable in clinical settings. Very recently, Nohwal and colleagues [75] fabricated a simple amperometric immunosensor for p53 detection as shown in Figure 8. A Pencil graphite electrode (PGE) was first dipped into piranha solution and then p53 antibodies were immobilized onto the electrode surface and was used to capture p53 antigen in the sample. The immunosensor exhibited a linear concentration range of 10 pg/mL to 10 ng/mL and LOD of 10 pg/mL and has other advantages such as fast response time and can be used for early cancer diagnosis. 

##### Label–Free EC Immunosensors for Cancer Antigen 15–3 (CA 15–3) Detection 

CA 153 is a glycoprotein belonging to mucin 1 family. It is present in healthy human beings at a level lower than 30 U/mL and is a major serum oncomarker for breast cancer used for both detection and follow–up [76,77]. It is also used in cases of ovarian cancer [16,78] as it’s level is heightened in about 70% of patients with epithelial ovarian cancer with most of the disease being in an advanced stage [78]. As such, it is crucial to develop sensitive analytical methods for the detection of CA153 at very low concentrations when the disease is at its early stage. Aiming to achieve this, Hasanzadeh et al. [79] used a GCE to design a label–free EC immunosensor for the detection of CA 153. The electrode surface was modified with thiolated graphene quantum dots (GQDs) and coated with gold nanospears (AuNSs) through electrochemical assembly before functionalizing with cysteamine (CysA) and subsequent treatment with EDC/NHS to enhance the immobilization of anti–CA 153. The CysA/AuNSs/GQDs hybrid interface provided a sufficient specific surface area for efficient CA 153 antibody immobilization and also increased bioactivity and stability of the immobilized antigens. FE–SEM and EDS photoelectron spectroscopies were used as the step–by–step process characterization tool for the sensor. For the electrochemical investigation and detection of CA 153, the group used SWV and CV. The sensor detected CA 153 concentration as low as 0.1 U/mL and linear range of 0.16–125 U/mL with good specificity in unprocessed human plasma samples. The sensor was also proposed for use to assay CA 153 in malignant cell line lysates. Khoshroo and teammates [80] in 2018 were able to detect CA 153 using a cobalt sulfide/graphene/AuNPs nanocomposite–fabricated label–free EC immunosensor. The group modified a graphite screen–printed electrode (SPE) with a suspension of CoS_2_–GR nanocomposite by drop drying. Then, Au NPs were electrodeposited onto the CoS_2_–GR/SPE surface followed by the immobilization of the anti– CA153. Au NPs functioned as an immobilization matrix for the antibodies. The nanocomposite of CoS_2_–GR enhanced the immobilization of the antibody due to its large surface area while also demonstrating great electrocatalytic properties against catechol oxidation. The established enzyme and label–free CoS_2_–GR–AuNPs/Ab/SPE immunosensor was used for CA 153 antigen detection in a wide linear range of 0.1–150 U/mL and LOD of 0.03 U/mL with good precision, specificity and reliability. In addition, Rebelo and other researchers [77] established a simple and rapid label–free EC immunosensor for the estimation of CA 153 concentration using gold screen–printed electrodes (AuSPEs). A self–assembled monolayer of mercaptosuccinic acid (MSA) was first formed on the electrode surface through covalent binding of the thiol group and the gold surface. Then, anti–CA 153 was coated onto the modified electrode surface using NHS/EDC technique to form anti–CA153/NHS/EDC/MSA/AuSPEs immunosensor. The highly selective fabricated immunosensor was then used to detect CA 153 in the range of 1.0 to 1000 U/mL with LOD of 0.95 U/mL. Again, a label–free EC immunosensor was designed by Amani et al. [76] for the quantification of CA 153 which exhibited an LOD of 0.3 U/mL and linear concentration range of 1.0 to 150 U/mL with a sensitivity of 1.88 µA (µM cm^2^). The trio used screen–printed graphite electrode which was functionalized with nanocomposite of reduced graphene oxide (rGO) and copper sulfide (CuS) to form CuS–RGO/SPE. Then, anti–CA 153 was immobilized onto the modified electrode after treating the electrode with 1–pyrenebutyric acid N–hydroxysuccinimide ester (PANHS). The detection was based on the electro–oxidation of catechol, which served as the electrochemical probe that increased current response. Immobilization of anti– CA 153 onto the modified electrode reduced response to catechol. And further immunocomplex formation by the antigen–antibody reaction, which covered the surface of the CuS–RGO/Ab/SPE and blocking the catalysis effect of CuS–RGO to the oxidation of catechol, which decreased the current response of the immunosensor.

Moreover, another group of researchers [81] designed an ultrasensitive label–free EC immunosensor using conductive dendritic Au@Pt core–shell nanocrystals (Au@Pt NCs) and ferrocene–grafted chitosan (Fcg–CS) for the detection of CA 153. CS was first conjugated with Fc through a Schiff base reaction, while Au@Pt NCs were prepared from their precursor– materials using a simple one–pot wet–chemical process in the presence of hexadecyldimethyl benzyl ammonium chloride as the growth–directing agent. Then, the prepared Au@Pt NCs was dispersed into the Fc–g–CS suspension under ultra–sonication to yield the Au@Pt NCs/Fc–g–CS nanocomposite which was used to modify a GCE through drop drying. Fcg–CS was used to improve stability and to provide large specific surface area for the attachment of Au@Pt NCs to enhance electrochemical signals and improve biocompatibility. Anti–CA 153 and BSA were then deposited in sequence onto the functionalized electrode to form BSA/Ab/Au@Pt NCs/Fc–g–CS/GCE immunosensor. Using the developed immunosensor to detect CA 153, an LOD of 0.17 U/mL and linear concentration range of 0.5–200 U/mL was reached with excellent stability, acceptable reproducibility and selectivity for CA15–3 and possible application in clinical determination of CA15–3 in serum samples.

##### Label–Free EC Immunosensors for CA 125 Detection

CA 125 (Mucin 16 or MUC 16) is a tumor biomarker mainly used for OC [82,83]. Though its level is also elevated in some other cancer types such as lung, endometrial and breast cancers [83]. But about 90% of OC patients are positive for CA125 [82,84]. CA125 assays can be utilized for early detection of ovarian cancer and to monitor the progress in OC patients as well as to distinguish between benign and malignant disease [85]. So it is important to develop sensitive and cost–effective diagnostic tools for its early detection to improve clinical outcomes. For this reason, Gasparotto et al. [84] proposed a highly sensitive label–free EC immunosensor for CA 125 using a gold–coated glass substrate modified with ZnO nanorods (ZnO NRs) that was prepared via assisted microwave hydrothermal method. Au NPs were also deposited onto the modified electrode by sputtering to form ZnO NRs–Au NPs nanohybrid. Then, the ZnONRs–AuNPs nanohybrid–functionalized electrode was coated with cystamine and glutaraldehyde as linkers in sequence to aid the immobilization of CA125 antibody. The anti–CA 125/ZnO NRs–Au NPs biosensor demonstrated a low LOD of 2.5 ng/µL. CA 125 has also been detected using a screen–printed paper–based EC immunosensor. The research team [83] first compounded reduced graphene oxide (rGO) and thionine (Thi) composite through π–π stacking interactions during the process of ultrasonic and vigorous mixing process. Then, AuNPs was mixed with the obtained rGO/Thi composite and by means of the interaction of the amino groups of rGO/Thi and AuNPs, rGO/Thi/AuNPs nanocomposites was formed. rGO/Thi/AuNPs nanocomposites were then used to functionalize a paper electrode and anti– CA 125 was immobilized for detecting CA 125 antigen. The immunoreaction of CA 125 antibody and CA 125 antigen forming an immunocomplex which reduced the current response of thionine was the applied detection principle. Hence, the reduced current signal was proportional to the corresponding CA 125 antigen concentration. Using DPV for electrochemical measurements, the linear range and detection limit achieved with the immunosensor was 0.1 U/mL to 200 U/mL and 0.01 U/mL, respectively. It also offered a good clinical application. Hasanzadeh et al. [82] devised another label–free EC immunosensing method for the detection of CA 125 utilizing a GCE. First, polydopamine (PDA) and electrochemically reduced graphene oxide (ERGO) were electrodeposited onto the bare electrode sequentially. The modified electrode was treated with cysteamine (Cys A) where its amine (NH_2_) groups bonded with the COOH groups of ERGO. Following, Au NPs were electrodeposited onto the functionalized electrode. The modified electrode was then used to immobilize HRP–labelled CA 125 antibody. The fabrication process was completed by rinsing off unbound antibody and blocking free sites with BSA. The immunosensor, anti–CA125/AuNPs/Cys A/ERGO/PDA/GCE was used to detect CA 125 antigen which resulted to an LOD of 0.1 U/mL and linear concentration range of 0.1 to 400 U/mL. Again, Zheng et al. [49] used a GCE to develop a label–free voltammetric immunosensor for CA125 detection. Platinum nanoparticles (Pt NPs) was used to modify polyaniline (PANI) hydrogel functionalized bare electrode. Then, Prussian Blue (PB) was grown on the surface of PtNPs followed by electrodeposition of Au NPs. PB–PtNPs were used as excellent conductive redox–active materials for the catalytic redox of H_2_O_2_ for signal enhancement, AuNP improved electrical conductivity and provided specific surface area to immobilize the antibody while PANI hydrogel possesses large specific surface for assembling the nanomaterials and has good conductivity which promotes electron transport. The fabricated Ab/AuNPs–PB–PtNP–PANI/GCE immunosensor had an LOD of 4.4 mU/mL and linear range of 0.01–5000 U/mL with sensitivity of 119.76 µA. (U/mL)^−1^ measured using SWV.

In 2018, Gazze and colleagues [86] established an EC immunosensor on a disposable graphene screen–printed electrode (SPE). An amine layer (Polyaniline (PANI)) was used to decorate the electrode to maintain the integrity of the graphene (Gr) and enable the immobilization of antibodies. Anti–CA 125 was deposited onto the surface of PANI/Gr–SPE through EDC/NHS activity and used to detect CA 125 antigen. The BSA/anti–CA125/PANI/Gr–SPE biosensor showed a wide linear range from 0.92 pg/μL–15.2 ng/μL and an LOD of 0.923 ng/μL. In addition, in 2020 Sangili et al. [87] proposed a simple label–free EC immunosensing technique for CA 125 detection using layer–by–layer AuNPs and reduced graphene–oxide (RGO) composites to modify a GCE (AuNPs/RGO/GCE). The modified electrode was decorated with self–assembly monolayer film of 11–mecaptoundecanoic acid (11–MUA). Using EDC/NHS chemistry, anti–CA125 was immobilized and used to detect CA 125 antigen. The as–prepared immunosensor exhibited a very wide linear concentration range of 0.0001 to 300 U/mL and LOD of 0.000042 U/mL with good stability, high selectivity, and reproducibility, and presented good results when used in patients’ samples. Recently, Gu and his team [88] described another EC immunosensing technique for the detection of CA 125 utilizing a microporous GCE. The microporocity of the GCE increased the surface area and improved its adsorption ability. Nanocomposites of nitrogen–doped reduced graphene oxide functionalized carboxylated multi–walled carbon nanotubes (N–rGO@CMWCNTs) were used to modify the microporous electrode. Then, EDC/NHS was used to activate the carboxyl groups on the surface of the modified electrode before coupling with highly conductive and biocompatible chitosan–decorated gold nanoparticles (CS@AuNPs) through a reaction between the carboxyl groups of CMWCNTs and amino and hydroxyl groups of chitosan. Anti–CA125 was immobilized onto the matrix for the detection of CA 125 antigen through formation of Au–S bonds with AuNPs using its –SH groups. The BSA/Anti–CA125/N–rGO@CMWCNTs/CS@AuNPs/GCE immunosensor was able to detect CA125 in the range of 0.1 pg mL^−1^–100 ng mL^−1^, with detection limit of 0.04 pg mL^−1^. In 2017, Paul et al. [85] used MWCNTs/ZnO (MZnONF) nanofiber composite as electroactive materials to modify a GCE for CA 125 detection. The nanofiber composite was synthesized by electrospinning method followed by calcination. The GCE/MZnONF platform was treated with EDC/NHS as crosslinking solution to activate –COOH groups on the surface of MZnONF nanofiber to enhance the attachment of CA 125 antibodies through its –NH2 groups. The label–free BSA/Anti–CA125/MZnONF/GCE electrochemical immunosensor showed a wide detection range of 0.001 U/mL to 1000 U/mL with a detection limit of 0.00113 U/mL and sensitivity of 90.14 µA/(U/mL)/cm^2^ with good stability, reproducibility, and acceptable selectivity. Figure 9 shows a very simple label–free EC immunosensor for the quantification of CA125 based on gold nanostructures (GNs) fabricated by a team of researchers [89]. GNs were casted on a gold–decorated glass wafer electrode via electrodeposition. Then, the modified electrode was treated with cysteamine and through EDC/NHS chemistry, anti–CA 125 was captured for the detection of CA 125 antigen. By using CV for electrochemical measurements, CA 125 antigen was detected in the range of 10 to 100 U/mL with LOD of 5.5 U/mL.

De Castro and partners [90] in 2020 developed another very simple and versatile label–free EC immunosensor which detected CA 125 in the range of 5 to 80 U/mL with LOD of 1.45 U/mL using DPV (Figure 10). A screen–printed graphite electrode (SPGE) was functionalized with the polymer, poly (3–hydroxyphenylacetic acid) (3–HPA) via electropolymerization and deposition followed by coating with anti–CA125 antibody and BSA treatment. The SPGE/poly(3–HPA)/anti–CA 125 immunosensor was then used to detect CA125 antigen in the presence of potassium ferrocyanide and KCL solutions. The biosensor showed good reproducibility, stability, and selectivity with a fast analysis time of about 30 min.

##### Label–Free EC Immunosensors for HER2 Detection 

As a receptor tyrosine kinase and a member of the epidermal growth–factor receptor (EGFR) family, HER2 participates in cell signaling and may cause growth, proliferation, differentiation and apoptosis of cells [91]. In most gynecological and other cancer types such as breast, ovarian, endometrial, stomach, lung, gastric, colorectal, pancreatic, bladder, colon, head, neck and salivary gland cancers, it is over–expressed and can be used for prognosis [91,92]. In a normal healthy human, the blood level of HER2 ranges from 2 to 15 mg/mL [31] and increases when there is anaplastic changes and tumor development [93]. Based on this, researchers have developed highly sensitive immunosensing techniques for HER2 quantification. For example, Nasrollahpour et al. [93] proposed a label–free EC immunosensor for HER2 detection with detection limit of 1 Fg/mL and wide linear concentration range of 1 ng/mL to 1 Fg/mL. The group first simultaneously electrochemically deposited WO_3_ and a polyglutamic acid (p–Glu) nano–biocomposite on a bare GCE. Then, anti–HER2 was treated with EDC/NHS solutions to activate the –COOH functional group on the antibodies before subsequent immobilization on the functionalized electrode to form the GCE–WO3/P–Glu–Ab immunosensor. The fabrication process was characterized using EIS, DPV and CV. The developed immunosensor exhibited excellent specificity, stability, and reproducibility. Augustine and his team [94] in 2019 suggested another rapid and ultrasensitive label–free EC immunosensing method for the quantification of HER2 as shown in Figure 11. The group first synthesized reduced graphene oxide (RGO) doped with molybdenum trioxide (MoO_3_) by in situ growth through one–pot low temperature hydrothermal synthesis to form MoO_3_@RGO nanohybrid. Then, the nanohybrid was functionalized with (3–aminopropyl) triethoxysilane (APTES) which is a linker agent. Through the process of electrophoretic deposition in the presence of a magnesium compound as the Charger salt, APTES/MoO_3_ @RGO nanohybrid was casted onto hydroxyl–decorated ITO electrode surface. Then, anti–HER2 coupled with EDC/NHS to activate −COOH groups of the antibodies was immobilized onto the modified electrode with the help of free amine terminals of APTES followed by spreading of BSA. The prepared BSA/anti–HER–2/APTES/MoO_3_@RGO/ITO immunosensor was used for HER2 detection and had a sensitivity of 13µA mL ng^−1^cm^−2^, with a linear concentration range of 0.001–500 ng/mL and LOD of 0.001 ng/mL determined using CV, DPV and EIS. The biosensor also possessed other features, such as excellent stability and reusability, high selectivity for HER2 and great potential for detecting HER–2 in the clinical samples.

Again, to detect HER2 using a label–free method, Ehzari et al. [95] made use of magnetite (Fe_3_O_4_) nanoparticles doped with TMU–21 through the use of a methacrylic acid (MAA) linker and decorated with carboxylated MWCNTs to modify a GCE as seen in Figure 12. HER2 antibody was captured on to the magnetic framework composite through EDC/NHS activity and the immunosensor was used to detect HER2 antigens. The reduction of H_2_O_2_ through the electrocatalytic activity of Fe_3_O_4_@TMU–21 and MWCNTs were used to determine the quantity of HER2 via amperometry. The linear concentration range, recorded with Fe_3_O_4_@TMU–21/MWCNTs/Ab immunosensor, was 1.0 pg/mL to 100 ng/mL with LOD of 0.3 pg/mL.

A label–free voltammetric immunosensing method was established by Wahyuni and colleagues [91] for the detection of HER2 utilizing a SPCE. The bare electrode was functionalized with electrodeposited AuNPs and coated with 3–mercaptopropionic acid (MPA) before immobilizing EDC/NHS, which enabled covalent binding with a HER2 antibody to form an SPCE/AuNPs/MPA/Ab immunosensor for HER2 antigen detection. The quantification of the antigen was based on the redox activity of ferricyanide/ferrocyanide electroactive species determined using cyclic voltammetry which realized a linear concentration range of 0 to 10 ng/mL and LOD of 2.9 ng/mL. Hartati et al. [96] proposed a label–free EC immunosensor for detecting HER2 based on antiHER2–AuNPs bioconjugates. The group sequentially mixed AuNPs, APTMS and PEG–NHS–Maleimide to get GNPs/APTMS/PEG–NHS–Mal solution. Then, thiolated anti–HER2 was added to the solution to form the bioconjugate GNPs/APTMS/PEG–NHS–Mal/anti–HER2 through the linker PEG–NHS–Mal. The bioconjugate was immobilized through amine coupling onto an SPCE–GNP electrode earlier coated with MPA and treated with EDC/NHS and cysteamine in sequence. The fabricated immunosensor SPCE–GNP/MPA/bioconjugate was then used to detect the HER2 antigen. Electrochemical measurements were carried out in the presence of ferricyanide redox pair using CV which yielded HER2 concentrations of 5.0 µg/mL and LOD of 1.20 × 10^−2^ ng/mL within a response time of 60 min.

##### Label–Free EC Immunosensors for Human Epididymis Protein 4 (HE4) Detection 

HE4 is a small molecular weight glycoprotein [97] belonging to the whey acidic four–disulfide core (WFDC) family and is over expressed in ovarian carcinomas [98] and is endorsed by The United States Food and Drug Administration (FDA) for monitoring relapse or progression of epithelial ovarian carcinoma [99]. It has higher sensitivity as a biomarker for OC for stage 1 disease than CA125 [100]. It also has high specificity for OC and can be used to detect early disease occurrence and recurrence [101]. When used together with CA125 it improves early detection and diagnosis of OC of any histological type or stage [102] and can be used as biomarker panel for detection and risk staging of OC [99,102,103] as specificity can increase to up to 100% [104]. Inspired by this, Yan and co–authors [101] fabricated a label–free EC immunosensor based on multi–amplification strategy for HE4 quantification. In the method, bimetallic palladium nanoarms doped gold nanorods (Au@Pd) holothurian–shaped NPs (Au@Pd HSs) were used to functionalize TiO_2_ nanoclusters marked nitrogen–labelled reduced graphene oxide (TiO_2_–rGO). This heterostructure (TiO_2_–rGO/Au@Pd HSs) was then used to modify a GCE via drop drying. Then, HE4 antibody was immobilized onto the functionalized electrode through physical adsorption followed by BSA. The heterostructures enhanced the PEC performance by providing excellent conductivity and electrocatalytic activity towards H_2_O_2_ reduction, and also offered good biocompatibility and large specific surface area for the biomolecules. The fabricated BSA/anti–HE4/TiO_2_ –rGO/Au@Pd HSs/GCE immunosensor was used to detect HE4 antigen in a linear concentration range of 40 fM/mL to 60 nM/mL and LOD of 13.33 fM/mL which were detected based on the reduction of H_2_O_2_. In addition, Qu and Yu [6] prepared a simple and low–cost voltammetric immunosensor on an FTO electrode for HE4 quantification. The team functionalized the bare electrode with a nitrogen–decorated graphene nanosheets–gold nanoparticles composite through electrodeposition to form (AuNPs/N–doped GNs/FTO). Then, they coated it with fructosyl amino–acid oxidase (FAO) using chitosan–glutaraldehyde crosslinking strategy before adding target HE4. The immunosensor exhibited high sensitivity in detecting HE4 with an LOD of 1.21 µg/mL and linear concentration of 10 to 65 µg/mL measured using DPV.

#### 2.1.2. Sandwich–Type EC Immunosensors for OC Biomarkers Detection

The sandwich–type biosensing strategy uses label materials and was developed as a result of the quest for highly sensitive detection methods. It involves the immobilization of the primary antibody on the surface of a modified electrode followed by incubation with the antigen of interest, and then the introduction of a labelled secondary antibody for the enhancement of signals. Although it is a complex system, the high sensitivity and selectivity, ease of fabrication, and rapid analytic time has made this method attractive [65,105,106]. 

##### Sandwich–Type EC Immunosensors for CEA Detection

Zhang and colleagues [107] in 2018 constructed a sandwich–type EC immunosensor for the determination of CEA. The group used a GCE functionalized with AuNPs (Au NPs/GCE) for Ab1 immobilization and CEA antigen recognition. Ag NPs and porous composite of microporous carbon spheres (CS) loading silver nanoparticles (Ag NPs) spaced hemin/reduced graphene oxide (AgNPs@CS–hemin/rGO) was used to label the detection antibodies (Ab2) AgNPs@CS–hemin/rGO–Ab2 and used as the sandwich. The irreversible stacking of rGO was overcome by CS loading Ag NPs (AgNPs@CS), which was used as a spacer insert in a hemin/rGO sheet, forming a porous structure with more active sites of hemin. The reduced graphene oxide served as a supporting material for hemin to prevent it from aggregation and oxidative self–destruction. Hemin is a peroxidase–like reduced H_2_O_2_ to amplify current signals. Ag NPs was also responsible for catalytic reduction of H_2_O_2_ that further elevated the current signal of the construct and also acted as biocompatible substrate with good electrical conductivity that promoted the loading of the capture antibodies (Ab1). The immunosensor exhibited good reproducibility, selectivity, and stability in a linear range of 20 fg/mL to 200 ng/mL and detection limit was determined to be 6.7 fg/mL. and 6.7 fg/mL as detection limit. Similarly, Xu et al. [108] also reported another sandwich type EC immunosensor for CEA quantification (Figure 13). The screen–printed carbon electrode (SPCE) used as the working electrode was modified with AuNPs–decorated polydopamines (Au/PDA), which served as substrate material onto which primary antibody (Ab1) was immobilized. Hydrothermally synthesized dendritic tri–fan blade–like PdAuCu nanoparticles (PdAuCu NPs) decorated with ferrocene–amine–doped graphene oxide (Fc–NH_2_–GO) were used to label the secondary antibodies (Ab_2_). The composite provided a large surface area that enhanced adequate adsorption as well as catalyzed the H_2_O_2_ reduction which amplified the current signal. In the presence of CEA, the bioconjugate Ab_2_/PdAuCu NPs/Fc–NH_2_–GO was used to form the sandwich. The proposed immunosensor presented a linear detection range from 0.1 pg mL^−1^ to 200 ng mL^−1^ and a limit of detection of 0.07 pg mL^−1^, with high sensitivity and excellent specificity towards CEA, good stability, reproducibility, and satisfactory precision. It also exhibited accurate and quantitative detection ability in human serum and urine samples. 

In addition, Nakhjavani et al. [109] reported the sandwich–type, ultra–sensitive and selective EC detection of CEA using a GCE modified with thiolated graphene oxide (T–GO) by the drop–casting method. Following, streptavidin–modified AuNPs were dropped onto the modified electrode. This platform provided an increased active surface area for the immobilization of the primary antibody for the capturing of CEA antigen. Streptavidin–coated silver nanoparticles (AgNPs) and horseradish peroxidase (HRP) composite were used to immobilize the biotinylated monoclonal antibody (secondary antibody) which served as the signaling probe to complete the sandwich. HRP was used to amplify the electrochemical signal through its catalytic electrochemical reduction of hydroquinone used as electron mediator in the presence of H_2_O_2_. The sensor attained a linear response range of 100 fg/mL to 5 pg/mL with a very low detection limit of 75 fg/mL. Lai and teammates [110] in 2019 proposed a sandwich–type EC immunosensor for detecting CEA. A GCE was used which was successively layered with electrochemically reduced graphene oxide (rGO) and with chitosan (CS) and Au NPs through drop drying method. This composite (rGO/CS/Au NPs) served as a platform for signal sensing to sufficiently capture the Ab1. Polythionine–gold (PTh–Au) composite synthesized by adopting a chemical reduction technique in one–pot was employed as the signal label for tagging Ab2 which significantly boosted the sensitivity of the developed biosensor by amplifying electrochemical signals. The–as prepared sensor had a linear detection range of 0.3 ng/mL to 30 ng/mL and detection limit of 0.1471 pg/mL. Similarly, Tian and his team [111] used a GCE to develop a sandwich–type EC immunosensor for the quantification of CEA. The team utilized reduced graphene oxide nanosheets functionalized with β–cyclodextrin (CD–NGs) as biocompatible platform with high electrical conductivity to immobilize the Ab1 for the biorecognition of CEA antigen. Nanocomposites of tri–metallic NiAuPt and reduced graphene oxide nanosheets (NGs) was used to label the Ab2 and used to form the sandwich in the presence of CEA antigen for electrochemical signal amplification. NiAuPt–NGs nanocomposites which showed very high electrocatalytic activity towards the reduction of hydrogen peroxide due to the synergetic effect present, was synthesized through one pot simultaneous reduction of GO and the metal precursors using NaBH_4_. A linear range of 0.001 to 100 ng/mL and ultralow LOD of 0.27 pg/mL was achieved with the fabricated immunosensor. Very recently Zhao et al. [112] proposed an ultrasensitive sandwich–type EC immunosensor for CEA quantification using a GCE. The group used the composite of highly conductive graphene oxide and biocompatible gold nanoparticles (GO–AuNPs) as the substrate for immobilization of the Ab1. Silver–platinum composite was capped with BSA and used as label for Ab2 (Ag@BSA–Pt/Ab2). The silver nanoparticles enhanced the electron transfer process while platinum improved electrochemical sensing through catalysis of hydrogen peroxide. The prepared immunosensor provided a linear detection range of 0.005 ng/mL to 100 ng/mL with detection limit of 0.76 pg/mL and excellent selectivity, suitability, stability and good reproducibility. Figure 14 shows a sandwich–type EC immunosensor developed by Jing et al. [113] to quantify CEA using a GCE. The group used 3–dimentional porous graphene oxide (3DHGO) to form nanocomposite with platinum using a wet chemical process. The prepared 3DPt/HGO nanocomposite served as the electroactive material used to modify the working electrode onto which the Ab1 was immobilized to capture CEA antigen. In the presence of the antigen, the Ab2 which was labeled with Au nanoparticles doped with Horseradish peroxidase was used to form the sandwich. Using CV, DPV and EIS, the electrochemical detection readings were realized to be 0.001–150 ng/mL and 0.0006 ng/mL for the linear detection range and LOD, respectively.

In 2018, Lv led a team [114] that used a GCE to develop another sandwich–type EC immunosensor for detecting CEA as shown in Figure 15. The electrode was coated with polydopamine modified with Au nanoparticles (Au@PDA). This matrix helped to speed up electron transfer process and functioned as a platform onto which Ab1 was immobilized with subsequent CEA antigen attachment. The Ab2 was tagged with nitrogen–doped graphene (NG) modified with copper ion (Cu^2+^) and functionalized with cubic Au doped with Pt dendritic nanomaterials (Au@PtDNs) to form Au@PtDNs/NG/Cu^2+^–Ab2. Au@Pt DNs efficiently enhanced the electrocatalysis of H_2_O_2_ reduction, Cu^2+^ promoted further redox of H_2_O_2_ to amplify the signal of the immunosensor while the good conductivity and large surface area of NG benefited the immobilization of large amount of Au@Pt DNs. This method produced a very sensitive and specific immunosensor for CEA detection with good reproducibility and stability and having an LOD of 0.167 pg/mL with 0.5 pg/mL to 50 ng/mL as the linear detection range. It also has great potential applications in clinical analysis for ultrasensitive detection of various tumor markers.

Li, Y. et al. [115] also constructed a sandwich–type EC immunosensor for the quantification of CEA which offered a linear concentration range of 0.1 pg/mL to 100 ng/mL and LOD of 0.0697 pg/mL. To develop such immunosensor, the team modified bare GCE through electrodeposition of gold nanoparticles. The Au NPs was used to immobilize the primary CEA antibody for CEA antigen capturing as well as to enhance electron transfer due to its properties such as high specific surface area, good biocompatibility and superior electrical conductivity. Meanwhile, amino treated magnetic graphene sheets stocked with gold and silver core–shell NPs to adsorb nickel ion and secondary CEA antibody was used as the probe to form the sandwich. The synergetic effect present in Au@Ag/Fe_3_O_4_–GS/Ni^2+^ nanocomposites increased the electrocatalytic reduction of H_2_O_2_ which bettered the sensitivity of the fabricated immunosensor. 

##### Sandwich–Type EC Immunosensors for CA 19–9 Detection

Based on the signal amplification associated with label use in sensing, Zhang et al. [116] developed a sandwich–type ultrasensitive amperometric immunosensor for CA19–9 detection using polydopamine–decorated silver NPs (PDA–Ag NPs) to label the secondary CA19–9 antibody. Graphene oxide– marked melamine (GO–MA) was used as the sensing platform onto which the primary CA19–9 antibody was immobilized for CA19–9 antigen capturing following modification with Au NPs. On introducing H_2_O_2_, Ag NPs were etched into silver ions through redox reaction which produced sharp peaks that were used to quantify the antigen concentration. A wide linear concentration ranges of 0.0001 to 100 U/mL and very low LOD of 0.032 mU/mL was reached. Using polyoxometalate doped AuNPs (AuNPs@POM) nanocomposite as immunosensing platform on a GCE, Yola and Atar [117] in 2021 fabricated another sandwich–type EC immunosensor for the detection of CA 199. Ab1 was immobilized onto the platform and used to capture CA199 antigen. After, through strong π–π and electrostatic interactions the signal antibody (Ab2) was tagged with the composites of 1D MoS_2_ nanorods/LiNb_3_O_8_ (1D MoS_2_ NRs/LNO–Ab2) and used to form the sandwich for signal amplification. The as–prepared sensor possessed good selectivity, stability and reusability with an LOD of 0.030 µU/mL and quantification limit of 0.10 µU/mL. 

##### Sandwich–Type EC Immunosensors for AFP Detection

Jiao and his group [14] proposed a sandwich–type EC immunosensor for AFP detection that displayed an ultra–low LOD of 0.05 pg/mL with wide linear range of between 0.1 pg/mL to 10 ng/mL. The group used a glassy carbon electrode functionalized with a polydopamine–modified N–doped multi–walled carbon nanotube (PDA–N–MWCNT) which enhanced electrochemical signals as the biocompatible platform for primary antibody immobilization. The secondary antibody was tagged with nanocomposite of amino group functionalized graphene–decorated Au@Pt mesoporous nanodendrites (NH_2_–GS/Au@Pt) and used as the probe. Graphene provided the surface area for sufficient attachment of highly electrocatalytically active and conductive Au@Pt nanodendrites which was employed to intensify signals. The sensor exhibited satisfactory selectivity, good reproducibility and stability and could be used in clinical analysis. Figure 16 shows a sandwich–type EC immunosensor proposed by some researchers [118] for detecting AFP in human serum using dual–metal composites. In the study, bimetallic Au and Ag NPs (Au@Ag) were used to functionalize polydopamine–decorated phenolic resin microporous carbon spheres (PDA–PR–MCS) which acted as the probe to label the secondary antibody (Ab2). The sensing platform for primary antibody (Ab1) immobilization was set by modifying the glassy carbon electrode surface with Au NPs through electrodeposition to provide a biocompatible and very conductive layer and to increase electron transfer rate. Electrochemical signals were enhanced on reduction of H_2_O_2_ by the probe (Au@Ag/PDA–PR–MCS–Ab2). The sensor had an LOD of 6.7 fg/mL and linear detection range of 20 fg/mL to 100 ng/mL. Another sandwich–type EC immunosensor was designed by Wei and colleagues [105] in 2016 which detected AFP within a linear range of 0.1 pg/mL to 50 ng/mL with an LOD of 0.033 pg/mL. The team modified a glassy carbon electrode with Au NPs through electrodeposition to form the platform onto which the primary AFP antibody (Ab1) was effectively immobilized for antigen–capturing and also to boost the electrochemical signals. The secondary antibody (Ab2) was tagged with graphene oxide and CeO_2_ mesoporous nanocomposite decorated with 3–amino propyltriethoxysilane doped palladium octahedral NPs (Pd/APTES–M–CeO_2_–GS). APTES–CeO_2_–GS nanocomposite provided large surface area for adequate capturing of palladium NPs which catalyzed H_2_O_2_ reduction used for electrochemical signaling, hence improving the sensitivity of the sensor. The stability and reproducibility of the sensor were good with excellent sensitivity and selectivity. Using the nanocomposite of graphene oxide/methylene blue/gold NPs (GO–MB–Au NPs) prepared through mixing and centrifugation method, Shen et al. [65] functionalized a glassy carbon electrode via a dip–coating technique for a sandwich–type EC detection of AFP. This platform served as the immunosensing matrix onto which AFP antibody was immobilized for the biorecognition of the antigen. The group utilized gold nanocubes (AuC) doped with horseradish peroxidase (HRP) to label the secondary antibody (AuC–HRP–anti–AFP) which was used to form the sandwich. The highly sensitive and selective immunosensor exhibited an LOD of 1.5 pg/mL with linear concentration range of 0.005–20 ng/mL which was determined using DPV.

##### Sandwich–Type EC Immunosensors for P53 Detection

Figure 17 shows a sandwich –type EC immunosensor proposed by Kang et al. [119] in 2020 for detecting p53 based on a signal amplification strategy. A GCE electrode was first functionalized with the conductive polymers; poly (3, 4–ethylenedioxythiophene): polystyrenesulfonate (PEDOT: PSS) using drop–drying method. Then, AuNPs were electrodeposited on the modified electrode and used for the immobilization of Ab1. The platform enhanced charge transfer and provided an ideal conducting surface while AuNPs provided large specific surface areas for Ab1 adsorption. The construct, Ab1/AuNPs/PEDOT: PSS/GCE was then used to capture p53 antigen. To prepare the redox probe, 2, 3–diaminophenazine (DAP) was encapsulated in ZIF–8, then Ab2 was conjugated to Zeolitic Imidazolate framework-8 (ZIF–8) using an NHS/EDC strategy to obtain ZIF–8–DAP–Ab2. The immunoprobe was then used to form a sandwich with the conducting platform in the presence of the antigen. The decomposition of ZIF–8 in the presence of an acidic solution (HCL) releases the electrochemical probe DAP, which amplifiers the signals and improves the sensitivity of the immunosensor. The developed immunosensor had a working range of 1–120 ng/mL and an LOD of 0.09 ng/mL with good recovery, high sensitivity, reliability, and selectivity.

##### Sandwich–Type EC Immunosensors for CA 153 Detection

Martins and team [120] recently introduced a sandwich–type EC immunosensing method for quantifying this enzyme in human serum as shown in Figure 18. The team functionalized an SPCE with layer–by–layer film of Au nanoparticles and reduced graphene oxide (Au–rGO) via co–electrodeposition to form the substrate for immobilization of primary anti–CA 153. Then, the secondary antibodies were labelled with horseradish peroxidase while hydroquinone was employed as an electron mediator. The developed system exhibited rapid detection of CA 153 with an ultra–low LOD of 0.08 fg/mL and a wide linear concentration range of 0.1 fg/mL to 1 µg/mL. It also had good selectivity for CA 153 as well as long–term storage at 4 °C for >30 days.

Another sandwich–type EC immunosensing method for detecting CA 153 was formulated by Nakhjavani et al. [121]. Gold electrode (GE) was first modified with streptavidin (Strp) and was used to immobilize biotinylated primary anti– CA 153 and BSA and used to capture CA 153 antigen to form GE/Strp/biotinylated mAb/CA15–3 platform for electrochemical sensing. Then, the probe which consisted of magnetic beads (MB) coated with streptavidin and functionalized with biotinylated HRP and linked to secondary biotinylated anti– CA 153 (biotinylated mAb/Strp–MB/biotinylated HRP) was added to form the sandwich for signal amplification. In the presence of hydroquinone (HQ) which was employed as the redox agent and H_2_O_2_ acting as HRP activating agent electrochemical measurements were taken. HRP catalyzes the reduction of HQ in the presence of H_2_O_2_ to further increase signals. The LOD realized with the immunosensor was 15 × 10^−6^ U/mL and linear concentration range of 50 to 15 × 10^−6^ U/mL using SWV. The immunosensor had excellent sensitivity and specificity with significant stability and also showed excellent detection potentials as compared to the results obtained using ELISA when used to detect CA 15–3 in human serum.

##### Sandwich–Type EC Immunosensors for CA125 Detection

A sandwich –like EC immunosensor was fabricated by Kumar et al. [122] for the quantification of CA 125 as shown in Figure 19. Capture CA 125 antibody was coated on an ITO electrode earlier modified with gold nanorods (Au NRs) and treated with glutaraldehyde which served as a linker. This platform, ITO–AuNRs–Ab served to capture CA 125 antigens. Then, the secondary CA 125 antibody was labelled with Au NPs and tagged with cadmium ion (Cd^2+^) to form a probe to generate more signals. In the presence of the antigen, the probe (AuNP–Ab–Cd^2+^) was introduced to form the sandwich and measurements were taken using DPV. The developed immunosensor was stable and reproducible and exhibited a detection limit of 3.4 U/mL and linear detection range of 20–100 U/mL.

Also, in 2020 Pakchin et al. [123] designed an EC immunosensor using a sandwich approach on a GCE for the detection of CA125. The group initially synthesized the composite of three–dimensional reduced graphene–oxide and multiwall carbon nanotubes (3DrGO–MWCNTs) through mixing and ultrasonication. The synthesized 3DrGO–MWCNTs composite was added to PAMAM/AuNPs composite prepared via reduction of HAuCl_4_ in the presence of PAMAM and continuous stirring. The mixture was ultrasonicated to yield 3DrGO–MWCNTs–PAMAM/AuNPs nanocomposites which were cast on the bare electrode. Using glutaraldehyde, anti–CA125 was immobilized. Polyamidoamine–modified gold nanoparticles enhanced conductivity and increased the specific surface area for adequate capture antibody attachment. The matrix, Ab–3DrGO–MWCNTs–PAMAM/AuNPs–GCE was then used to capture CA125 antigen. Magnetic nanoparticles doped with O–Succinyl–chitosan (Suc–Cs@MNPs) and modified with toluidine blue (TB) were used to label the secondary antibody to form Ab–Suc–Cs@MNPs–TB which served as the tracer. In the presence of the antigen, the tracer was used to form a sandwich. Using SWV for electrochemical measurements, a wide linear range of 0.0005–75 U/mL with an LOD of 6 µU/mL was attained. The immunosensor was also very stable, selective, sensitive and reproducible. Pakchin led the same group [124] in 2018 that fabricated another sandwich–type EC immunosensor for CA 125 detection. The group modified the electrode with the bionanocomposites of chitosan–AuNPs/MWCNT/GO which served as the substrate for CA 125 primary antibody immobilization and attachment of the antigen. Then, AuNPs–marked lactate oxidase (AuNP–LOx) was used to label the secondary antibody and employed as the probe for the sensor. Electrochemical measurements were based on the oxidation peaks of hydrogen peroxide catalyzed by chitosan–AuNP which functioned like peroxidase. The linear concentration range were observed to be 0.01 to 100 U/mL with and LOD of 0.002 U/mL using chronoamperometry. The immunosensor was comparable to ELISA with excellent reproducibility, selectivity, and stability. Again, a sandwich–type voltametric–immunosensing assay has been described by Iyer et al. [125] for the detection of CA125 using basically, nitrogen–doped reduced graphene oxide (NrGO) nanocomposites together with thionine (Thi) and AuNPs. The researchers prepared composites of NrGO with carbon nanofibers (CNF) and NrGO with carbon nanotubes (CNT) and mixed them each with Thi, PBS and glutaraldehyde in sequence to form the transducing materials followed by addition of Ab1, BSA and AuNPs. The mixtures were coated on bare GCE for the capturing of the antigen. The same nanomaterials and method were used for the signal antibody (Ab2) which was used to form the sandwich in the presence of the antigen. A wide concentration ranges of 10 U mL^–1^ to 32 × 10^–4^ U mL^−1^ was reached with satisfactory linearity observed on using NrGO/CNT and NrGO/CNF while NrGO/CNF showed better sensitivity with an LOD of 0.28 U mL^−1^.

##### Sandwich–Type EC Immunosensors for HER2 Detection

In 2018, Shamsipur et al. [126] constructed a sandwich–type EC immunosensor for HER2 detection with excellent sensitivity. Through a simple co–precipitation process, Fe_3_O_4_ (Magnetite) NPs were first prepared. Then, 3–aminopropyltrimethoxysilane (APTMS) was used to functionalize it via an ultrasonic assisted synthesis. Followed by the introduction of aminated anti–HER2 through glutaraldehyde linker to form the detection probe antiHER2/APTMS–Fe_3_O_4_ which was used to modify a GCE. In the presence of HER2 antigens on the bioconjugate platform, the label probe was used to form a sandwich. The label probe was synthesized by adding APTMS–Fe_2_O_4_ to AuNPs before functionalizing with hydrazine and then adding thiolated anti–HER2 to it forming Hyd@AuNPs–APTMS–Fe_3_O_4_–labeled anti–HER2. Hydrazine reduced Silver ions on the surface of AuNPs to silver. Then, the deposited silvers were stripped. The stripping signals were used for immunosensing which was done by DPV. A linear concentration ranges of 5.0 × 10^−4^ to 50.0 ng/mL and LOD of 2.0 × 10^−5^ ng/mL was achieved. The sensor was applied to serum samples and results were in line with those of ELISA, hence could be used in clinical settings. In addition, Ahmad and colleagues [92] suggested a facile sandwich–type EC immunosensing technique for determining HER2 based on the signal amplification of lead sulfide QDs–bound Ab2 (Ab2–PbS QDs). PbS QDs was reacted with carbonyldiimidazole linker to enable the attachment of Ab2. The formed Ab2–PbS QDs was then used as label. To prepare the immunosensing platform, an SPCE was electrochemically pre–treated with H_2_SO_4_ to form carboxyl functional groups on the surface which was activated using EDC/NHS. Then, followed by sequential immobilization of primary anti–HER2 (Ab1) and BSA and subsequent HER2 antigen recognition. Following, the Ab2–PbS QDs label was dropped onto SPCE/Ab1/HER2 to form the sandwich. The sensing performance of the sensor was determined by the stripping signal of Pb(II) brought about by acid dissolution. Using SWV for electrochemical measurements, a linear concentration ranges of 1 to 100 ng/mL and LOD of 0.28 ng/mL was detected with a recovery of 91.3% to 104.3% in spiked samples. In 2021, Ma [127] single–handedly constructed another sandwich–type EC immunosensor based on the use of AuNPs grown on MnO_2_ nanosheets (MnO_2_ NSs/AuNPs) for the quantification of HER2. This nanocomposite was used to modify the working electrode and used as the electrochemical platform onto which primary HER2 antibody was immobilized for the capturing of HER2 antigen. To form a sandwich, Au@Ag nanorods (Au@Ag NRs) was employed for dual signal amplification and used to label the Ab2. Au@Ag nanorods catalyzed the reduction of H_2_O_2_ which amplified the current signal and subsequently favoured the transfer of electrons between Ag and Ag^+^. Using DPV and chronoamperometry, HER2 antigen was quantified with linear concentration range from 50 fg/mL to 100 ng/mL and 100 fg/mL to 100 ng/mL with LOD of 16.7 and 33.3 fg/mL, respectively. In addition, Yola [128] alone developed a sensitive sandwich–type voltammetric immunosensor to detect HER2 based on the use of AuNPs doped with Cu organic framework (AuNPs/Cu–MOF) as the matrix to decorate a GCE. Following modification of the GCE, anti–HER2–Ab1 was immobilized through amino–gold interaction and used to capture HER2 antigen to form HER2/anti–HER2–Ab1/AuNPs/Cu–MOF/GCE platform. Then, the composite of Cu_2_ZnSnS_4_ nanoparticle (CZTS NP) quaternary chalcogenide and platinum (Pt)–doped g–C_3_N_4_ (Pt/g–C_3_N_4_) prepared in a one–pot hydrothermal system, was used to label anti–HER2–Ab2 to form the signaling probe CZTS NPs/Pt/g–C_3_N_4_/anti–HER2–Ab2 for signal amplification. The probe was used to form the sandwich on the immunosensing platform in the presence of the antigen. The detection of the antigen was based on the oxidation of H_2_O_2_ (used as the Redox probe) to O_2_ through the co–electrocatalysis of the quaternary chalcogenide and platinum NPs. Additionally, the quaternary chalcogenide and platinum NPs synergistically enhanced signals through provision of high specific surface area and enhanced charge separation. The immunosensor showed high selectivity, reproducibility, stability, and reusability with a linear concentration range of 0.01 and 1.00 pg mL^−1^ and LOD of 3.00 fg mL^−1^. Very recently, Safari [129] developed a sandwich–type EC immunosensor for detecting HER2. The researcher employed the magnetic framework, Fe_3_O_4_@TMU–24 which was decorated with AuNPs and functionalized with mercaptoacetic acid (MAA) monolayer (Fe_3_O_4_@TMU–24–AuNPs/MAA) as the sensing materials to modify a GCE. The carboxyl groups on the surface of MAA was activated by EDC/NHS to enable immobilization of Ab1 for the capturing of HER2 antigen. In the presence of the antigen, the Ab2 tagged with palatine (Pt) doped CdTe QDs (Pt: CdTe QDs–Ab2) was launched as the label signal to form a sandwich. Pt: CdTe QDs possess excellent biocompatibility and catalytic activity towards H_2_O_2_ redox which enhances signals, hence making the biosensor to achieve a wide detection range from 1 pg/mL to 100 ng/mL and LOD of 0.175 pg/mL. The sensor also exhibited long stability, good reproducibility and specificity and can be used to analyze clinical samples.

##### Sandwich–Type EC Immunosensors for HE4 Detection

Yan et al. [130] developed a sandwich –type EC immunosensor for the detection of HE4 based on the use of amine–functionalized graphene doped gold nanorods (AuNRs/NH_2_–GS) as the sensing matrix for adequate immobilization of HE4 Ab1. AuNRs/NH_2_–GS composite was first used to modify a GCE electrode before coating it with HE4 Ab1 and BSA. The modified electrode (BSA/Ab1/Au NRs/NH_2_ –GS/GCE) was then used to capture HE4 antigen. The NH_2_–GS increased the surface area and conductivity of the electrode and Au NRs enabled the immobilization of the Ab1 and also efficiently quickened the rate of electron transfer. Core shell Au–doped Pd urchin shaped nanostructures (Au@Pd USs) were used to label the Ab2 (Au@PdUSs–Ab2) and to catalyze the reduction of H_2_O_2_ for signal amplification. In the presence of the antigen on the sensing platform, Au@PdUSs–Ab2 was immobilized to form the sandwich. The fabricated immunosensor had and LOD of 0.33 pmol/L with linear concentration range of 1 pmol/L to 50 nmol/L. Interestingly, Fan, et al. [131] used nanodendrites of multimetallic AgPtCo (AgPtCo NDs) produced using one–pot method as signal amplifier and redox probe to construct an ultrasensitive sandwich–type EC immunosensor for the detection of HE4. AgPtCo NDs has large surface area onto which HE4 Ab2 were immobilized. The group prepared the magnetic nanocomposites, Fe_3_O_4_@SiO_2_@Au MNCs (M@SiO_2_@Au MNCs) and modified it with HE4 Ab1 and used it as magnetic capture probes (MCPs) to enrich and separate HE4 from the sample with the assistance of an additional magnet. Then, AgPtCo NDs–tagged Ab2 were added to MCPs–HE4 to form the sandwich complex. The sandwich complex was then immobilized onto a magnetic glassy carbon electrode (MGCE). DPV was used to carry out electrochemical measurements. The developed biosensor showed a wide linear concentration range of 0.001–50 ng/mL with an LOD of 0.487 pg/mL and agrees with standard methods when used on real samples. It also had satisfactory specificity, good reproducibility and exceptional stability. In 2018, Cadkova and group [132] constructed another sandwich–type EC immunosensor for HE4 detection using SPCE. The group coupled HE4 antibodies to CdSe–ZnS QDs nanocomposite which was used to form a sandwich with magnetic microparticles doped with HE4 antibodies that captured HE4 antigen. Two configurations of metal film–modified screen–printed carbon electrodes were used, mercury film electrode (Hg–SPCE), and bismuth–film electrode (Bi–SPCE). The Magnetic particles coupled– immunocomplex anti–HE4 IgG–HE4–anti–HE4 IgG CdSe/ZnS was immobilized on each of the functionalized electrode. Quantification of HE4 was done with Square wave anodic stripping voltammetry (SWASV) and is based on current response of Cd^2+^ and its release from CdSe/ZnS by acid hydrolysis at the electrodes. The linear concentration ranges were found to be 20 pM to 40 nM and 100 pM to 2 nM with LOD of 12 pM and 89 pM for the Hg–SPCE and Bi–SPCE, respectively. The immunosensor also possesses the right sensitivity and specificity required for routine clinical diagnostic.

### 2.2. PEC Immunosensors for OC Biomarkers Detection

#### 2.2.1. PEC Immunosensors for CEA Detection

As discussed earlier, PEC sensing offers better results than ordinary electrochemical sensing due to the presence of light which leads to enhanced signals and reduced background noise. The immobilization platforms are usually photoactive materials which can be excited by photons of suitable energy; and innovative engineering of these materials has led to significant improvements in the development of PEC sensors. For instance, the use of heterojunction photocatalysts (formed between a semiconductor and another semiconductor, carbon or metals) [127]. Heterostructured photocatalysts have been found to exhibit better photocatalytic properties than their separate components and are very effective in charge separations and transfer in the visible light region, leading to a higher photoelectric conversion efficiency [128,129]. Wu and colleagues [133] in 2018 fabricated a highly selective, sensitive, and satisfactorily stable label–free PEC immunosensor for CEA quantification exploiting the advantages of heterojunction formed between Zn_0_._1_ Cd_0_._9_S/g–C_3_N_4_ nanocomposites. The nanocomposites were prepared using an in–situ growth procedure with a hydrothermal method and showed excellent photoelectrochemical activity compared to the pure Zn0.1Cd0.9S and g–C_3_N_4_. The nanocomposite was used to modify an ITO electrode through drop drying. Using mecaptoacetic acid and EDC/NHS chemistry, CEA antibody was immobilized for subsequent CEA antigen detection. An LOD of 1.4 pg/mL with wide linear detection range from 0.005 ng/mL to 20 ng/mL was achieved with the immunoassay protocol providing a promising platform for clinical diagnostics. Fan et al. [134] fabricated an ultrahigh sensitivity sandwich–type PEC immunosensor for the detection of CEA which exhibited a linear range of 0.5 pg/mL to 100 ng/mL and an LOD of 0.16 pg/mL. The group first used TiO_2_ to functionalize a bare ITO electrode. Then, TiO_2_ was sensitized by modification with CdSeTe alloyed QDs through electrostatic adsorption assisted by oppositely charged polyelectrolyte. The following were coated with CdS: Mn shells via successive adsorption and reaction of Cd^2+^/Mn^2+^ and S^2−^ to form TiO_2_/CdSeTe@CdS: Mn nanocomposites which served as the photoelectrochemical matrix for immobilization of capture anti–CEA (Ab1). Chitosan was used as a linking molecule to immobilize Ab1. CuS nanocrystals were used to label the signal anti–CEA (Ab2–CuS) and used for signal amplification during antigen–antibody recognition reaction. The Ab2–CuS conjugate synergistically reduced the photogenerated current because of the competitive light absorption and consumption of the electron donor by the photocatalytic properties of CuS nanocrystals and the steric hindrance caused by the signal antibody. The established immunosensor demonstrated good reproducibility, specificity and stability. Again, Nie and Co–workers [135] constructed a label–free PEC immunosensor for CEA detection using nanocomposites of poly (5–formylindole) and electrochemically reduced graphene oxide (P5FLn/erGO) to modify an ITO electrode. First, graphene oxide was electrochemically deposited on the electrode, next P5FLn was coated on it through electrochemical polymerization before introducing Au NPs by drop drying as shown in Figure 20. The nanocomposite acted as the photo active material as well as electroactive mediator. Au NPs was used to crosslink anti–CEA with the photo active materials. The synergy between P5FLn and erGO increased the photo current signals when irradiated with visible light but decreased upon anti–CEA immobilization. The photo current signal further decreased with the recognition event between anti–CEA and CEA. The proposed immunosensor had high specificity, good stability and reproducibility with a wide linear response range from 0.0005 to 50 ng mL^–1^ and low detection limit of 0.14 pg mL^–1^ and also gave satisfactory results when used to detect CEA in real human serum samples.

Zhang, B. et al. [136] constructed a visible light driven label–free PEC immunosensor using WO_3_/Au/CdS nanocomposite as shown in Figure 21. WO_3_/Au nanocomposite were first prepared on an ITO electrode by reversible charging and discharging mechanism based on the electrochromism nature of WO_3_, then a layer of CdS was coated on the surface of WO_3_/Au nanocomposite using successive ionic layer adsorption and reaction method. Following, the modified electrode was treated with TGA which interacted with CdS to immobilize –COOH functional groups then EDC/NHS was used to activate –COOH groups for anti–CEA immobilization. The as prepared WO_3_/Au/CdS composite was used as the photocatalyst. The separation of the photogenerated electron–hole pairs and the visible light absorption was greatly enhanced by the localized Surface Plasmon Resonance effect of AuNPs in conjunction with the photosensitization of CdS. The increase in photocurrent signals was up to 218 fold of WO_3_ and 87 fold of CdS using a light source of 430 nm. CEA detection was based on the antigen–antibody recognition and the change in photocurrent intensity was linearly related to the logarithm of CEA concentration. The linear range was 0.01 to 10 ng/mL with detection limit of 1 pg/mL. It had good stability, repeatability and reproducibility, and applicable to serum samples. 

Hu et al. [137] proposed a simple label–free PEC immunosensor for CEA quantification using Au NPs/WS_2_ nanocomposites. First, WS_2_ crystals were exfoliated to form WS_2_ nanosheets through ultrasonication in N–methyl–pyrrolidinone while AuNPs were made in the same pot via in–situ reduction of chloroauric acid. Then, AuNPs/WS_2_ nanocomposites were formed through self–assembled interaction of AuNPs with WS_2_ nanosheets. The formed Au/WS_2_ nanocomposite was then used to functionalize a GCE surface by drop–coating before immobilization of CEA antibody and subsequent treatment with glutaraldehyde which was used as a crosslinker. Following, BSA was dropped onto the modified electrode to block unspecific binding sites. The prepared immunosensor, anti–CEA(BSA)/Au/WS_2_/GCE was used to detect CEA antigen in the range of 0.001 to 40 ng/mL with detection limit of 0.5 pg/mL. It also exhibited good sensitivity, selectivity, and accuracy for CEA determination and was successfully used for CEA determination in clinical serum samples. In 2016, Li et al. [138] established a sandwich type PEC immunosensor to detect CEA based on the signal amplification of melamine network decorated with CdSe QDs which was used as label for Ab2. The group used TiO_2_ doped with Au NPs as the substrate to functionalize an ITO electrode onto which the Ab1 was attached to capture the CEA antigen. In the presence of the antigen, the label Ab2–CdSe/melamine was immobilized onto the modified electrode to form the sandwich. The immunosensor exhibited a very wide linear detection range of 0.005 to 1000 ng/mL with LOD of 5 pg/mL. It also showed excellent specificity for CEA with acceptable reproducibility and stability. The proposed method could provide a promising immunoassay for CEA detection in real samples and can be used as a platform for other protein determinations. Song’s team [139] prepared a sandwich–type PEC immunosensor for CEA detection based on the use of the heterojunction Z–type photocatalyst, MoS_2_/g–C_3_N_4_–PtCu nanocomposites as label to tag Ab2. The alloyed PtCu NPs can collect electrons and improve the electron transport efficiency hence improving the sensitivity of the sensor towards CEA detection. While g–C_3_N_4_ minimizes aggregation of PtCu NPs and MoS_2_, the heterojunction formed between MoS_2_ and g–C_3_N_4_ can enhance photoelectrocatalysis significantly. To form the platform for immunorecognition, the group modified a GCE with AuNPs and coated it with the capture antibody (Ab1) and BSA successively before CEA capturing. In the presence of CEA antigen, the probe was used to modify the surface of the electrode to form a sandwich. The proposed immunosensor achieved a linear concentration range of 0.0001 to 80 ng/mL and an LOD of 33 fg/mL with recovery rate of 103–104%. It also had good reproducibility, stability, and selectivity.

Figure 22 shows a more complex but ultrasensitive sandwich–type PEC immunosensor for CEA detection based on nanocomposites of CdS nanorods and bimetallic PdPt nanozyme for CEA determination described by Chen et al. [140]. Firstly, CdS nanorods were prepared through a solvothermal process while PdPt mesoporous spheres were synthesized using a one–step reduction method. Through a simple ultrasonic mixing, CdS/PdPt nanocomposites was formed and used to modify an ITO electrode to get CdS/PdPt/ITO. Then, Ab2 was conjugated with ZIF–8@GOx to form ZIF–8@GOx–Ab2. Ab1 was allowed to react with CEA antigens of various concentrations in a 96–well microplate followed by the addition of ZIF–8@GOx–Ab2 solution to form a sandwich. The obtained solution was then dropped onto the modified electrode. The peroxidase–like behavior of PdPt boosted the photogenerated current through catalyzing the bio–etching of CdS and subsequent generation of insoluble precipitate that changes the photocurrent signals. In addition, through its localized surface plasmon resonance effect and heterojunction formation with the CdS electron, hole pair separation and charge transfer processes were accelerated, which increased photocurrent signals. PEC measurements were taken in the presence of Na_2_SO_4_ which acted as the hole scavenger and 500W xenon lamp while chronoamperometry was used for electrochemical measurements. The fabricated PEC biosensor exhibited a wide linear response range of 1−5000 pg/mL with a very low detection limit of 0.21 pg/mL with high stability, selectivity, and potential in detecting CEA in human serum.

Also, Zhang and co–workers [27] developed a sandwich–type PEC immunoassay for CEA using C_3_N_4_–BiOCl semiconductor on a PDDA treated ITO electrode under visible light illumination utilizing L–cysteine as a sacrificial electron donor (Figure 23). The group first put CEA antibody and BSA in sequence into a 96–well plate. After incubation, the bioconjugate CuO–Ab2 used as the label was added into it. Then, following an acid treatment to release Cu^2+^ from CuO nanolabel, the solution was used to coat the C_3_N_4_–BiOCl/PDDA/ITO electrode for PEC detection. The Cu^2+^ released from CuO after treatment with acid, combined with both the cysteine and the electron receptors of C_3_N_4_ and BiOCl which decreased the photocurrent response which is inversely proportional to the CEA concentrations. The realized detection linear range with the sensor is 0.1 pg/mL to 10 ng/mL and detection limit of 0.1 pg/mL with satisfactory selectivity, repeatability and stability.

In 2016, Wang et al. [141] reported an ultrasensitive label–free PEC immunosensor for the detection of CEA which was developed with 2D TiO_2_ nanosheets and carboxylated graphitic carbon nitride (g–C_3_N_4_) as photoactive materials. A prepared solution of carboxylated g–C_3_N_4_/TiO_2_ nanocomposite was used to cover the surface of ITO electrode via drop–drying and sintering. CEA antibody and BSA were then coated onto the modified electrode surface sequentially, where anti–CEA bounded to TiO_2_ through the dentate bond formed between its carboxyl group and TiO_2_. The developed immunosensor was then used to detect CEA antigen at a voltage of 0.1 V and light intensity of 150 W/cm^2^ and in the presence of ascorbic acid as the sacrificial electron donor within a detection linear range of 0.01 to 10 ng/mL and LOD of 2.1 pg/mL. The immunosensor also exhibited high sensitivity, good selectivity, and stability with potential use in other analysis. In addition, Han and co–authors [142] proposed a label–free PEC immunosensor for the detection of CEA using CdS nanowires sensitized with (tungsten trioxide) WO_3_ and bismuth oxyiodide (BiOI) heterostructure on an ITO electrode. First, the WO_3_@BiOI heterostructure was used to modify an ITO electrode. Then, the functionalized electrode was coated with CdS nanowires via sequential chemical bath deposition method done by dipping the modified electrode interchangeably into solutions of [Cd(NH_3_)_4_]^2+^ and S^2–^ many times and capped with TGA. The formed ITO/WO_3_@BiOI@CdS electrode served as an immobilization platform for CEA antibody. After immobilizing antibodies through EDC/NHS chemistry, the immunosensor was utilized to detect CEA antigen in the range of 0.01 ng/mL to 50 ng/mL with a detection limit of 3.2 pg/mL at a bias voltage of 0 V and light intensity of 180·W·m^−2^. It showed good stability, high sensitivity and reproducibility and gave a satisfactory result when applied to human serum sample. In 2019, a highly selective, reproducible and stable sandwich–type PEC immunosensor for determining CEA was engineered by Wang and his team [28] as shown in Figure 24. The group used Cu–doped TiO_2_ sensitized with carbon nitride (Cu: TiO_2_/g–C_3_N_4_) which served as the photosensitizing materials for photocurrent enhancement to functionalize an ITO electrode (Cu: TiO_2_/g–C_3_N_4_/ITO). Following, CEA–Ab1 through EDC/NHS activity was immobilized with BSA to make BSA/CEA–Ab1/Cu: TiO2/g–C_3_N_4_/ITO electrode which was used for CEA antigen capturing. Then, alkaline phosphatase (ALP), as specific enzyme tags, was used to form the ALP–Au–Ab2 label used for the sandwich for signal amplification. ALP was employed for the in–situ generation of ascorbic acid from ascorbic acid 2–phosphate. The ascorbic acid was the electron donor used to scavenge the holes generated so as to further enhance the photocurrent signal. And the detection of CEA was based on the photocurrent signals corresponding to the concentration of the generated electron donor, which is directly proportional to the quantity of CEA antigen. A linear detection range of 0.005–1000 ng/mL and detection limit of 1 pg/mL was realised.

Another sandwich–type PEC Immunosensor for the detection of CEA was proposed by Liu et al. [143] using an L–like GCE. The electrode was functionalized with two–dimensional rhenium disulfite nanosheets (2D–ReS_2_) as an excellent photoactive material with large specific surface area where the CEA Ab1 was immobilized to form the anti–CEA(BSA)/ReS_2_/GCE immunosensing platform for CEA antigen detection. Then, CEA Ab2 labelled with alkaline phosphatase (ALP–Ab2) was also immobilized to form the sandwich. Ascorbic acid which was produced in situ through alkaline phosphatase catalysis of vitamin C magnesium phosphate, served as an electron donor to get rid of the photogenerated holes from ReS_2_ to produce photocurrent signals used for CEA detection. The developed simple immunosensor has a detection linear concentration range of 0.0005 to 10.0 ng/mL and LOD of 0.468 pg/mL with superior stability, selectivity and accuracy. In addition, Xie et al. [32] developed a sandwich–type PEC immunosensor for quantifying CEA. Gold–decorated TiO_2_ nanorods were used as signal promoter to coat an FTO electrode (Au@TiO_2_/FTO) followed by immobilization of CEA Ab_1_ through EDC/NHS coupling with subsequent blocking of non–specific binding sites with BSA. Then, the modified electrode was used to capture CEA antigen. To quantify the captured CEA antigen, CdSe nanospheres sensitized with BiVO_4_ was used as signal generator to label CEA Ab_2_ (CdSe@BiVO_4_–Ab2) through physical absorption and then used to form the sandwich. TiO_2_ provided large surface area and has good biocompatibility for the antibodies. The marching energy levels between TiO_2_, the CdSe and BiVO_4_ sufficiently utilized the light energy, which effectively enhanced charge separation and improved the PEC performance. The developed immunosensor possesses high sensitivity, good reproducibility and long–term stability with detection linear range of 0.01 ng/mL to 50 ng/mL and LOD of 0.5 Pg/mL. In 2021, Liu et al. [144] introduced a label–free PEC immunosensor for the detection of CEA using an FTO electrode. The group first used poly dimethyl diallyl ammonium chloride (PDDA) to cover the electrode followed by assembling of g–C_3_N_4_/CdSe nanocomposites as the photosensitizers onto the modified electrode. Then, the electrode was treated with EDC/NHS to activate –NH_2_ and –COOH groups on the surface to enable immobilization of anti–CEA. BSA was utilized to block non–specified binding sites on the modified electrode for CEA antigen detection. The as–developed biosensor showed excellent selectivity and repeatability with detection linear range of 10 ng/mL to 100 µg/mL and LOD of 0.21 ng/mL. 

#### 2.2.2. PEC Immunosensors for AFP Detection

Zang and co–workers [145] constructed a sandwich–type PEC immunosensor for AFP quantification using WS_2_ nanosheets doped with CdS QDs (WS_2_/CdS) to modify an ITO electrode. The modified electrode was used to immobilize the primary anti–AFP (Ab1) for the capturing of AFP antigen. Having the AFP captured, the secondary anti–AFP (Ab2) labelled with HRP (HRP–Ab2) was used as a sandwich, serving as a signal indicator. HRP biocatalyzed the reduction of 4–chloro–1–naphthol to nonconductive precipitate of benzo–4–chlorohexadienone in the presence of H_2_O_2_ as oxidant. The precipitate blocked electron transfer from the electron donor, hence causing more reduction in photocurrent signals. The PEC sensor was able to detect AFP in the range of 1 pg/mL to 20 ng/mL with LOD of 0.43 pg/mL and showed excellent anti–interference ability, satisfactory stability and acceptable reproducibility with desirable potential application in clinical settings. In addition, Wu et al. [146] developed a sandwich–type ratiometric PEC immunosensor which detected AFP in the range of 0.4 pg/mL to 40 ng/mL with an LOD of 0.2 pg/mL using TiO_2_ NTs sensitized with Carboxylated g–C_3_N_4_ as photoactive materials. This nanocomposite photoactive material was used to functionalize the electrode. Then, primary antibody (Ab1) was introduced onto the modified electrode with the aid of EDC/NHS chemistry and used to capture AFP antigen which reduced the photocurrent signal. To form a sandwich in the presence of the antigen, the secondary antibody (Ab2) which was capped with Co_3_O_4_ NPs was immobilized on the modified electrode for signal enhancement through steric hindrance of Ab2 and electron donor consumption of Co_3_O_4_ NPs. The as– prepared biosensor exhibited good selectivity and long term stability. In addition, Han et al. [147] proposed a label–free PEC immunosensor for the detection of AFP. The team decorated an ITO electrode with ZnO flower rods and then electrodeposited AuNPs on to the modified electrode forming Au–ZnO flower–rods heterostructure (Au–ZnO FRs electrode). The functionalized electrode was incubated with AFP antibody and BSA in sequence to form the immunosensor BSA/Ab/Au–ZnO. The AuNPs employed enhanced the visible light absorption of the ZnO flower rods as a result of the surface plasmon resonance effect, as well as boosted the photo–generated electron–hole pair separation which improved the PEC performance. The fabricated label free immunosensor was used to detect AFP antigen within a linear range of 0.005 ng/mL to 50 ng/mL and a detection limit of 0.56 pg/mL. The immunosensor also has other features such as high selectivity for AFP, acceptable stability, sensitivity, efficient and cost–effective. For the improvement of sensitivity, selectivity and better accuracy, a signal–on sandwich–type PEC immunosensor for the detection of AFP was established by Chen and Zhao [148] using CdSe QDs as the photoactive material on an ITO electrode. The modified electrode was functionalized with chitosan and glutaraldehyde as crosslinkers to immobilize AFP with subsequent BSA treatment. Then, Biotinylated AFP antibody (Bio–anti–AFP) used as detecting probe was captured onto the surface of the electrode forming antigen– antibody immunocomplex. Following, streptavidin (SA) which serves as the signal capturing agent and biotin–capped apoferritin encapsulated ascorbic acid (Bio–APOAA) which is the source of the sacrificial electron donor (ascorbic acid–AA) that prevents electron–hole recombination were immobilized in sequence to form ITO/CdSe/AFP/BSA/Bio–anti–AFP/SA/Bio–APOAA which was subjected to PEC measurements. The sensing strategy was based on the in–situ enzymatic hydrolysis of Bio–APOAA to release the sacrificial electron donor, AA, which enhances the photocurrent generation. The immunosensor exhibited a detection linear range of 0.001 to 1000 ng/mL and detection limit of 0.31 pg/mL with high selectivity and good sensitivity. Using a fluorine–doped tin oxide electrode (FTO), a simple label–free PEC immunosensor was fabricated by Jiang et al. [149] for the determination of AFP (Figure 25). The electrode was modified with sol–gel–prepared microporous ZnO inverse opals structure before coating with Ag_2_S nanoparticles synthesized via successive ionic layer adsorption and reaction. The sensitization improved charge separation and caused a subsequent increase in photocurrent generation. AFP antibody was immobilized onto the functionalized electrode through chitosan linkage before covering it with BSA (FTO/ZnO/Ag_2_S/CS/anti–AFP/BSA). The developed immunosensor was used to detect AFP and then subjected to PEC measurements. A wide concentration linear range of 0.05 ng/mL to 200 ng/mL with LOD of 8 pg/mL was realized. The immunosensor demonstrated good sensitivity and repeatability.

Recently, Li et al. [150] designed a label–free PEC immunosensor for the detection of AFP using Cesium tungsten bronze (Cs_x_WO_3_) and Au heterogeneous films. An FTO electrode was first treated with H_2_O_2_/H_2_SO_4_ to obtain a hydrophilic surface for the self–assembly of PMMA. Then, a gold film was formed on the functionalized electrode surface. Following, Cs_x_WO_3_ prepared through a colloidal synthesis was spin–coated onto the FTO/Au electrode surface. Then, AFP antibody was immobilized through chitosan–glutaraldehyde linkage and used to capture AFP antigen after modifying with BSA. The surface Plasmon resonance of Au and great photoelectric performance of Cs_x_WO3 enhances the photoelectric response of the sensor by improving the spectral absorption range and promoting charge transfer which significantly intensifies the photocurrent response. The developed FTO/AuCs_x_WO_3_/CS/Ab/BSA/AFP biosensor was subjected to PEC measurements and a detection linear range of 0.01 ng/mL to 500 ng/mL and detection limit of 7 pg/mL was obtained with excellent specificity, repeatability and long–term stability. Moreover, another team [151] proposed a highly selective sandwich–type PEC immunosensor for the quantification of AFP exploring the use of all–carbon nanohybrid of fullerenes (AC60)/graphite flakes (Gr)/graphene oxide (GO), (AC60–Gr–GO) as photoactive materials for signal amplification. AC60–Gr–GO nanohybrid was synthesized via mechanical mixing and sonication. Then, mixed with EDC/NHS, the label antibody (Ab2) and BSA in sequence to form Ab2@AC60–Gr–GO bioconjugate. Then, an ITO electrode was coated with GO, and EDC/NHS in sequence before the immobilization of the capture antibody (Ab1). The modified electrode was used to capture AFP antigen. In the presence of the antigen, the label bioconjugate was mounted to form the sandwich and PEC measurements were taken. The carbon nanohybrids greatly enhanced the photocurrent and also provided rich specific surface areas for bioconjugation. The immunosensor ITO/GO/Ab1/BSA/AFP/Ab2@AC60–Gr–GO demonstrated a wide linear detection range of 1 pg/mL to 100 ng/mL and an LOD of 0.54 pg/mL with excellent reproducibility and stability and applicable in clinical analysis.

#### 2.2.3. PEC Immunosensors for CA 125 Detection

In 2019, Xue led a group of researchers [152] to propose a signal–off PEC immunosensor for the detection of CA125 using core–shell NPs of SiO_2_ doped with polydopamine (SiO_2_@PDA) as quencher. The quencher was later used to tag the Ab2. Next, composite of CdTe QDs and chitosan (CS) was used to modify an ITO electrode then Ab1 was immobilized onto the CdTe–CS/ITO electrode through a crosslinking reaction with glutaraldehyde. The electrode was subsequently incubated with CA125 antigen. With the antigen captured, the signal–off quencher Ab2–SiO_2_@PDA was dropped onto the surface to form a sandwich. SiO_2_ NPs enhanced the uniform loading of PDA while promoting charge transfer impedance signal–off. PDA which has wide absorption in UV–vis region with excellent fluorescence quenching capability together with the photoelectric interaction of CdTe QDs used as the photoactive probe made the sensor to be highly sensitive with an ultra–low LOD of 0.3 mU/mL and linear concentration range of 1 mU/mL to 100 U/mL. Ge and colleagues [153] used an internal light source to develop a sandwich –type PEC immunosensor for the detection of CA 125. The team used reduced graphene oxide (RGO) to modify a paper working electrode (PWE) through in–situ reduction of GO followed by coating with zinc nanorods (ZNRs) and electrodeposition of CdS to form CdS/ZNRs/RGO–PWE photoactive sensing platform. Then, TGA was used to cover the functionalized electrode, while EDC/NHS were added to activate the COOH groups on the TGA–decorated electrode surface to enable attachment of Ab1 and CA 125 antigen capture. Following, the composites of N–aminobuthyl–N–ethylisoluminol (ABEI), GO and HRP (ABEI–GO@HRP) was used to label Ab2 and utilized as the probe to form the sandwich in the presence of the antigen. ABEI served as the chemiluminescent material to generate the internal light source through HRP catalyzed oxidation by H_2_O_2_. CdS improved the photo–to–electric conversion efficiency of the system while ZNRs/RGO–PWE provided large specific surface area for adequate deposition of CdS. The developed sensor, ABEI–GO@HRP/Ab2/Ag/Ab1/CdS/ZNRs/RGO–PWE possessed good specificity, stability and reproducibility with linear concentration range of 5.0 × 10^−4^ U/mL to 500 U/mL and LOD of 2.0 × 10^−4^ U/mL.

#### 2.2.4. PEC Immunosensors for CA 199 Detection

Zhu et al. [154] developed a sandwich–type PEC immunosensor for CA 19–9 detection utilizing Au NPs functionalized TiO_2_ nanowires modified with CdSe doped ZnS quantum dots as photoactive materials. The formed nanocomposite was used to modify an ITO electrode using layer–by–layer technique as shown in Figure 26. The formed layer, TiO_2_ NWs/Au/CdSe@ZnS served as the photoelectrochemical platform and used for capture CA 19–9 antibodies (Ab1) immobilization through EDC coupling. The signal antibodies (Ab2) were labelled with N–(2– carboxymethyl)–N′–methyl–4,4′–bipyridinium (V^2+^) and used for signal amplification which was achieved through its quenching effect due to the strong electron–withdrawing capability caused by the electron deficiency in its structure. Additionally, with the steric hindrance of the Ab2, photocurrent signals were well decreased. The PEC sensor showed a low detection limit of 0.0039 U/mL and wide linear range of 0.01 to 200 U/mL with acceptable reproducibility, satisfactory specificity and good storage stability.

#### 2.2.5. PEC Immunosensors for HE4 Detection

Zhang and colleagues [155] developed a PEC immunosensor for the detection of HE4 based on inorganic/organic heterojunction. The group first functionalized an ITO electrode with the composites of WO_3_ nanoplates and AuNPs (ITO/WO_3_/Au) prepared by reversible redox reaction. Then, PDA was coated onto the modified electrode via oxidative self–polymerization of dopamine to form ITO/WO_3_/Au/PDA. Through the Michael reaction between amino groups of the antibody and Quinone groups of PDA, HE4 Ab was immobilized and used to capture HE4 antigen. The immunosensor was then subjected to PEC measurements. PDA sensitized the photocatalyst, WO_3_ to broaden its visible light absorption. The synergistic effects of the heterojunction formed between WO_2_ and PDA and the plasmon resonance energy transfer effect of Au aided electron–hole separation, which effectively increased photocurrent response. The proposed PEC sensor offered a wide linear concentration range from 0.01 ng/mL to 200 ng/mL and LOD of 1.56 pg/mL with excellent reproducibility, specificity, and stability.

Photoanodic and photocathodic sensors which are the two major types of PEC systems have been noted to have high sensitivity and low selectivity, low sensitivity and good selectivity, respectively [156]. However, photoanodes and photocathodes can be used together to form a system where the former serves to amplify signals and the later used as the sensing platform for the analyte, hence producing a better system [156]. Capitalizing on this, a novel self–powered cathodic PEC immunosensing method was introduced in 2020 by Zhang et al. [156] for detecting HE4 as shown in Figure 27. The team used an ITO glass coated with the composites of WO_3_ and Au (ITO/WO_3_/Au) as the counter and reference electrode representing the photoanode for the PEC system, while another ITO electrode functionalized with the composites of Bi self–doped Bi_2_WO_6_ (Bi_2 + x_WO_6_) with p–n homojunction serving as the photocathode. The photocathode was further coated with polydopamine (PDA) to allow the attachment of HE4 Ab and subsequent detection of HE4 antigen. The ITO/Bi_2+x_WO_6_/Ab/BSA/HE4 electrode was then subjected to PEC measurements in the presence of H_2_O_2_ as the electron acceptor/donor. The PEC immunosensor demonstrated excellent performance with an LOD of 77.8 fg/mL and wide linear range of 0.001 to 100 ng/mL with satisfactory stability, excellent selectivity, reproducibility, and applicability in real serum samples. This great performance is attributed to the dual photoelectrode used.

Nanobodies also called stable single domain antibodies [157] have high specificity and affinity for antigens as compared to antibodies due to their smaller molecular weight of 12 to 15 kDa and can be used as recognition elements in biosensing [97]. Their smaller size increases their solubility and stability, facilitates production, and makes them to have low steric hindrance to reach targets [158]. Motivated by the above, Chen et al. [97] ventured into the establishment of a signal–on PEC sandwich–type immunosensor for the detection of HE4. The group used solvothermal synthesized–porphyrin metal–organic framework nanosphere (nPCN–224) coupled with the signal nanobody (Nb3) as the PEC probe for a signal–on. A nanocomposite of MWCNTs–PDA–AuNPs (M–P–A) was used as the substrate to modify an ITO electrode, which was used to anchor the capture nanobody (Nb2) through EDC/NHS activity for the immobilization of HE4 antigen. nPCN–224 served as the photoactive material with good optical properties and photoelectric conversion efficiency and excellent biocompatibility. In addition, its very ordered porosity and high surface area favors electron transfer. Dissolved O_2_ was employed to enhance charge separation and transfer during photoelectric conversion while ascorbic acid was used to scavenge the O_2_^–^ generated during the process which boosted the photocurrent signal. As a result of these, the Nb3@nPCN–224/Ag/Nb2/M–P–A/ITO immunosensor had a wide detection range of 1.00 pg/mL to10.0 ng/mL with a low detection limit of 0.560 pg/mL. The team reported that it could distinguish different stages of ovarian cancer from healthy people and that the result obtained matched well with those of electrochemiluminescence immunoassay. Most interestingly, Wei et al. [159] recently described a simple label–free PEC–EC dual mode immunosensing technique for the detection of HE4 using composites of AuNPs and CdS nanosheets (AuNPs/CdS NSs). The composite was applied onto an FTO electrode through a two–step electrodeposition followed by sequential immobilization of HE4 antibody and BSA and successive HE4 capturing via immunorecognition. The surface plasmon resonance effect of AuNPs improved the photoelectric characteristics of CdS by enhancing electron transfer which in turn bettered the PEC performance. PEC measurements were taken in the presence of ascorbic acid using a bias voltage of 0 V while DPV was used for EC measurements. The sensor possessed excellent specificity and stability with potentials for use in clinical settings. The dual mode biosensor demonstrated a resultant wide linear concentration range of 0.01 to 200 ng/mL and LOD of 1.084 pg/mL. 

## 3. Multiplex Immunosensors

Multiplexed systems involves the use of multiple transducers and/or detection tags that independently act on different specific analyte [160]. It quickens clinical diagnosis, as multiple biomarkers are being analyzed in a single test [161]. They have extra advantages like, improved detection efficiency of as low as femtograms per milliliter, versatility, reliability, good precision and reproducibility, smaller sample volume and short analysis time [160,161]. It is also more informative about the patient [161]. Propelled by this, Kuntamung et al. [162] constructed a simple label–free multiplex EC immunosensor for the simultaneous quantification of mucin 1, CA 153 and HER2 using an SPCE with 3–working electrode (WE) configuration. Each WE was functionalized with polyethyleneimine doped–gold nanoparticles (PEI–AuNPs). Then, followed by immobilization of the respective antibodies labelled with specific redox active species; anthraquinone–2–carboxylic acid (AQ) for MUC 1, thionine chloride (TH) for CA153, and AgNO_3_ (Ag^+^) for HER2 antibodies. The different responses of the redox species during the antigen–antibody immunoreaction were used for the multiplex sensing to quantify the antigens through measuring the changes in oxidation peak currents. The resulted decreased currents which are logarithmically corresponding to the antigen concentrations are in the range of 0.10 to 100 U/mL for CA153 and 0.10 to 100 ng/mL for MUC1 and HER2 while LOD was recorded to be 0.21 U/mL, 0.53 ng/mL and 0.50 ng/mL, respectively. Another simple label–free multiplex EC immunosensor was proposed by Cotchim and colleagues [163] for the concurrent detection of CEA, CA153 and CA 125. Graphene (Gr) was first used to functionalize ITO coated– glass working electrode scratched into three equal parts. Then, each part was decorated using methylene blue–chitosan (MB–Chi) composite (MB–Chi/GR/ITO). Then, glutaraldehyde was used as a crosslinker to immobilize the anti–CEA, anti–CA153 and anti–CA125 onto the separate modified electrodes for subsequent antigen detection. Gr increased electron transfer rate and the surface area of the electrodes. The fabrication process was monitored using CV, EIS and scanning electron microscopy. The linear concentration ranges and LOD achieved with the developed immunosensor are 0.10 to 100.00 pg/mL and 0.04 pg/mL for CEA and 0.10 to 100.00 mU/mL and 0.04 mU/mL for both CA153 and CA125 with good accuracy, reproducibility, and selectivity. Alizadeh et al. [164] fabricated a simple sandwich–type magnetoimmunosensor for the quantification of CEA and AFP antigens simultaneously using two different nanotags. Firstly, the group modified an ITO electrode with magnet which adsorbed Fe_3_O_4_ tagged Ab1 (Fe_3_O_4_–Ab1) which was used to capture CEA and AFP antigen and utilized as the electrochemical platform. Then, thionine doped 4–amiothiophenol (4ATP) functionalized AuNPs (thionine–4ATP–Au NP) and ferrocene–decorated 4–amiothiophenol (4ATP)–coated AuNPs (ferrocene–4ATP–Au NPs) were used as identifiable tags for CEA and AFP secondary antibody (Ab2), respectively, with thionine and ferrocene as the electroactive agents. Where 4ATP was used as the general linker. And for the coupling of CEA, glutaraldehyde was used to bind thionine to 4ATP. For the AFP, EDC/NHS chemistry was utilized to attach ferrocene to 4ATP through the interaction between the amino group of 4ATP and the carboxylic group of ferrocenes. The formed anti–AFP/ferrocene–4–ATP–Au NPs and anti–CEA/thionine–4–ATP–Au NP bioconjugates were immobilized onto the electrochemical platform to form a sandwich. Based on the redox response of the electroactive agents, thionine and ferrocene, detection of CEA and AFP was successful. The easy–to–fabricate ITO/Fe_3_O_4_–anti–CEA, AFP–CEA, AFP–AbCEA–4ATP Au–thionine/AbAFP–4ATP –Au –ferrocene immunosensor possessed good sensitivity, acceptable stability and repeatability with linear concentration ranges and LOD of 0.05–120 ngmL^−1^, 0.018 ngmL^−1^ and 0.05–100 ngmL^−1^, 0.012 ngmL^−1^ for CEA and AFP, respectively. Another multiplex immunosensor was proposed in 2016 for detecting CEA, CA 199, CA125 and CA 242 [165] (Figure 28). Metal ions of Cu, Pb, Cd and Zn were separately doped with chitosan–poly (acrylic acid) nanospheres (CP) and used to prepare the immunosensing probes with anti–CEA (Cu–CP–anti–CEA), anti– CA199 (Pb–CP–anti–CA199), anti–CA 125 (Cd–CP–anti–CA125) and anti–CA 242 (Zn–CP–anti–CA242). Then, chitosan–decorated AuNPs (CHIT–AuNPs) composites prepared via microwave assisted pyrolysis was used to coat a pretreated GCE. The functionalized electrode was dipped into a mixed solution of the capture antibodies, through glutaraldehyde linkage, the antibodies were immobilized and used to capture their corresponding antigens. Following, the immunosensing probes were mounted onto the immunosensing platform to form a sandwich–type by incubating the modified electrode in a mixed solution of the probes. Electrochemical measurements were taken using the anodic peaks produced by the metal ions which corresponded to the concentration of the various antigens. The realized linear concentrations with the proposed immunosensor were from 0.1 to 100 ng/mL for CEA, 1 to 150 U/mL for CA199, CA125 and CA242, while the LODs were 0.02 ng/mL, 0.4 U/mL, 0.3 U/mL and 0.4 U/mL, respectively. The immunosensor possesses other advantages such as good stability, specificity, and repeatability.

Table 1 contains a summary of the various EC and PEC immunosensors that have been recently developed for the determination of ovarian cancer biomarkers.

## 4. Conclusions and Future Perspectives

A review of recently developed electrochemical and photoelectrochemical immunosensors for detecting ovarian cancer biomarkers was provided. Our discussion focused mainly on the major biomarkers of ovarian cancer and selected papers published within the past seven years. Herein, EC and PEC immunosensors for detecting CEA, CA125, CA 153, CA 199, HE4, p53, HER and AFP were discussed. 

The electrochemical immunoassays were a combination of sandwich–type and label–free, with no obvious difference in the detection sensitivity. In some cases, depending on the composition of the electrochemical sensing platform and the probe used, a very low detection limit is achieved. For the photoelectrochemical immunosensors, the choice of photoactive materials and method of immobilization had a great impact on the photoelectric conversion efficiency, sensitivity, and stability of the sensors. Some factors pose great challenges in the development and usage of EC and PEC immunosensors. For instance, the instability of immunosensors has been a lingering issue. The reported immunosensors, so far, have a short shelf life, hence affecting their efficiency and limiting their approval for clinical usage. Developing better strategies to immobilize materials and biomolecules through stronger covalent bonding might go a long way towards solving this problem. In addition, using novel materials that do not deteriorate can have a positive impact on the half–life of biosensors. Second, the sensitivity of the sensors is a problem, as most immunosensors fail to detect OC biomarkers at a very low concentration for early–stage disease when treatment is possible. Careful choice of sensing materials can be helpful in improving the efficiency and sensitivity of immunosensors. The lack of reusability is another factor, as reusability can significantly save on cost. Simple methods of fabrication should be developed to make immunosensors reusable. Certain reagents may also be used for washing the immunosensor after use that will maintain the integrity of the sensor. In addition, multiplex EC and PEC immunosensors are important for OC detection, as more than one biomarker is used for more accurate and specific diagnosis. Unfortunately, very few studies have been conducted on multiplex EC, while none exists for multiplex PEC.

## Figures and Tables

**Figure 1 sensors-23-04106-f001:**
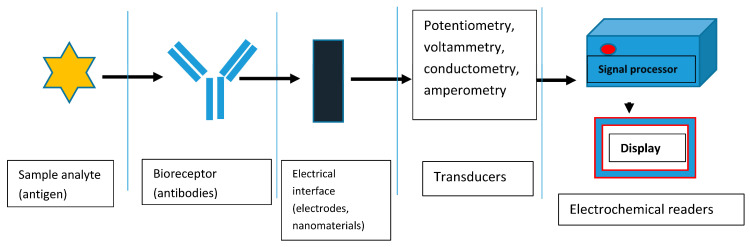
Basic components of a typical EC immunosensor.

**Figure 2 sensors-23-04106-f002:**
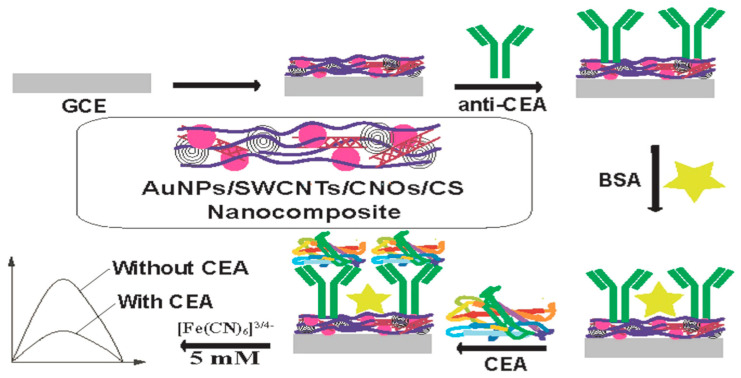
Construction steps of AuNPs/CNOs/SWCTNs/CS/anti–CEA biosensor. SWCNTs: single–walled carbon nanotubes; CNOs: carbon nano–onions; CS: chitosan; BSA: Bovine serum albumin. Reprinted with permission from [42], published by Elsevier, 2018.

**Figure 3 sensors-23-04106-f003:**
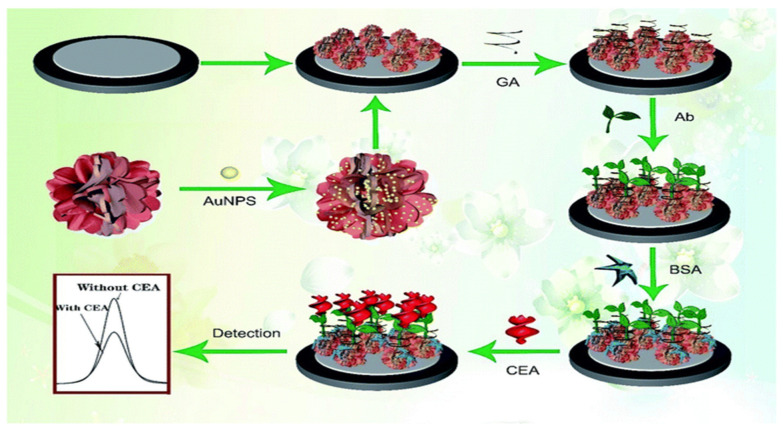
Schematic illustration of the fabrication procedure for CEA immunosensor. GA: glutaraldehyde; Ab: antibody; BSA: bovine serum albumin. Reprinted with permission from [43], published by The Royal Society of Chemistry, 2020.

**Figure 4 sensors-23-04106-f004:**
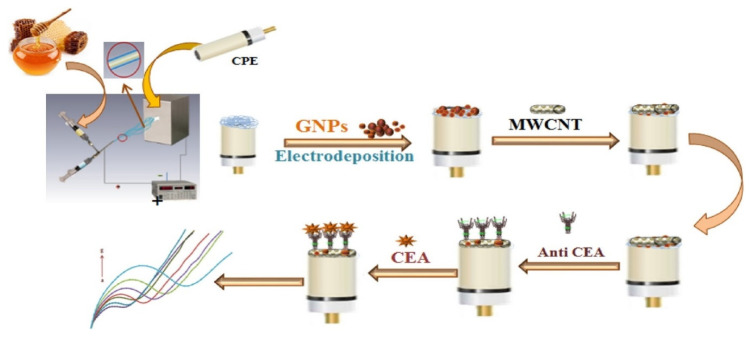
Electrospinning of core–shell honey nanofibers (HNF) onto the surface of a carbon–paste electrode (CPE) and preparation procedure of MWCNTs/GNPs/HNF/CPE based biosensor. MWCNTs: multi wall carbon nanotubes; GNPs: gold nanoparticles. Reprinted with permission from [46], published by Elsevier, 2020.

**Figure 5 sensors-23-04106-f005:**
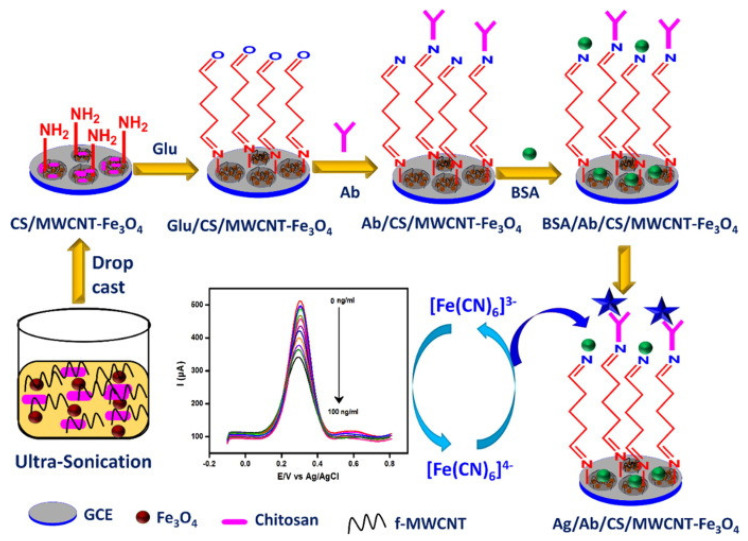
Schematic illustration of the fabrication process of CS–MWCNT–Fe_3_O_4_ based immunosensor for electrochemical detection of CA19–9 antigen. CS: chitosan; MWCNT: multiwalled– carbon nanotube; Fe_3_O_4_: magnetite nanoparticles; Ab: antibody; Ag: antigen. Reprinted with permission from [55], published by Elsevier, 2021.

**Figure 6 sensors-23-04106-f006:**
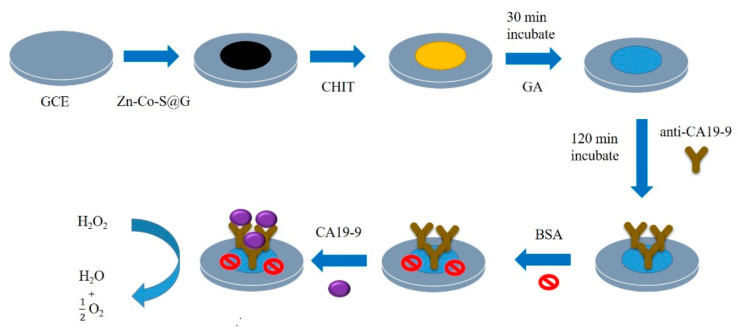
Fabrication process of electrochemical immunosensor for CA 19–9 detection. CHIT: chitosan; GA: glutaraldehyde; BSA: bovine serum albumin; Zn-Co-S@ G: binary transition metal sulfide (Zn-Co-S) doped with graphene (G) [53].

**Figure 7 sensors-23-04106-f007:**
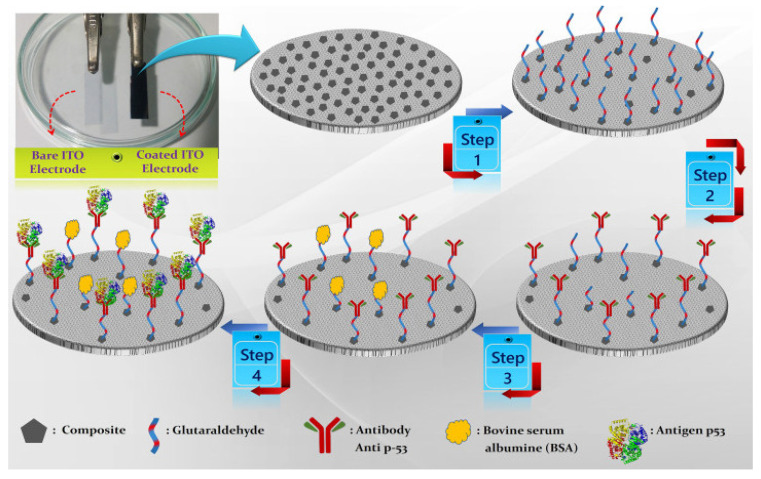
The schematic representation of chitosan and carbon black composite based electrochemical immunosensor. Step 1: modification of electrode with glutaraldehyde; step 2: immobilization of antibody; step 3: blocking with BSA; step 4: antigen capturing. Reprinted with permission from [69], published by Elsevier, 2018.

**Figure 8 sensors-23-04106-f008:**
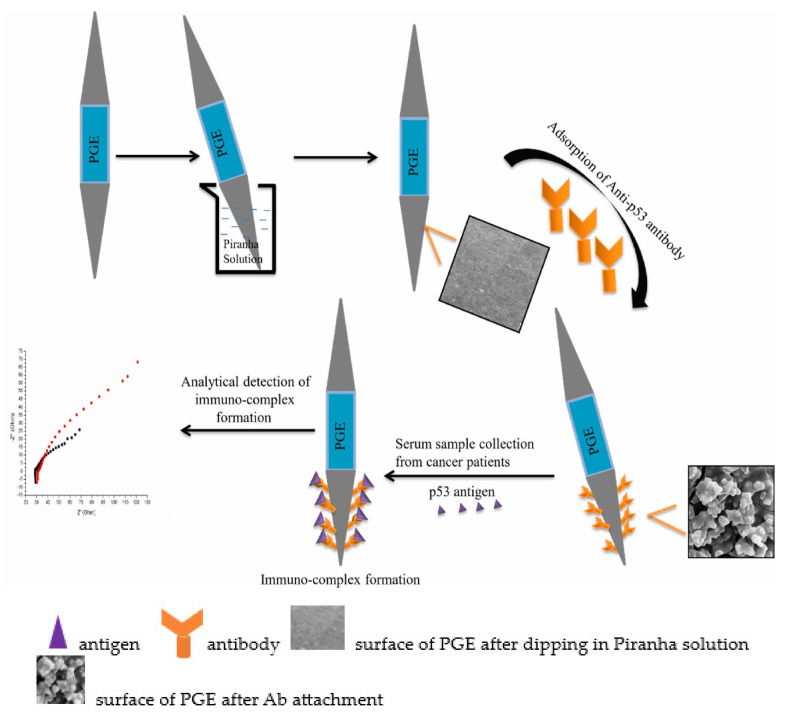
Steps in the fabrication of EC immunosensor. PGE: Pencil graphite electrode; Ab: antibody. Reprinted with permission from [75], published by Elsevier, 2021.

**Figure 9 sensors-23-04106-f009:**
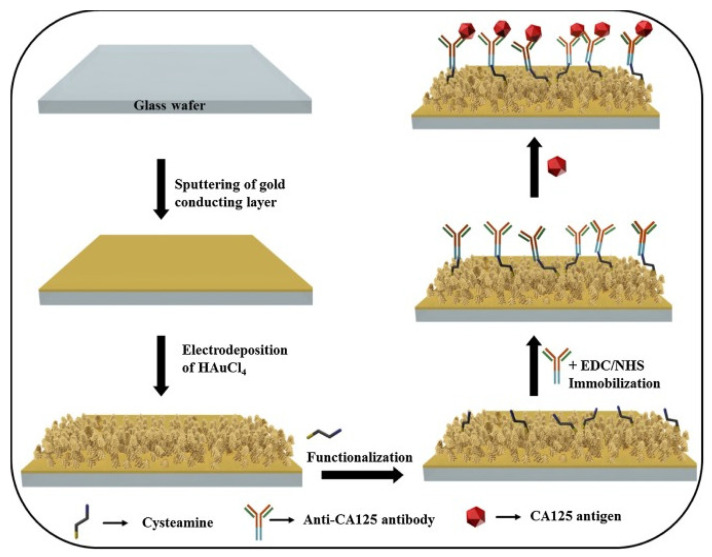
Step–by–step fabrication of electrochemical immunosensor. Reprinted with permission from [89], published by Elsevier, 2016.

**Figure 10 sensors-23-04106-f010:**
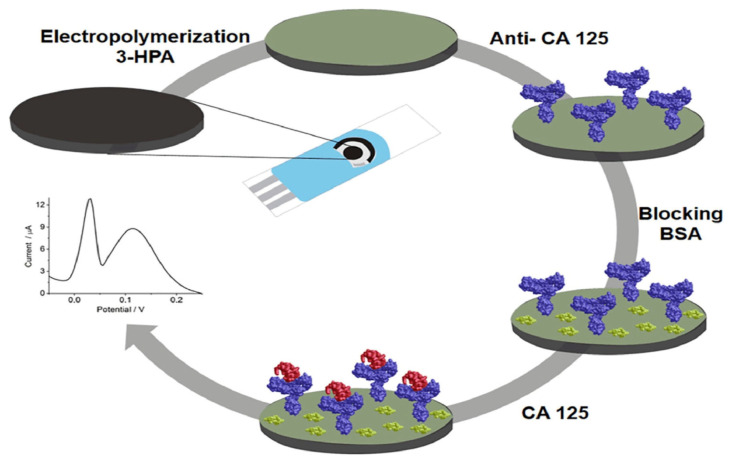
SPGE/poly(3–HPA)/anti–CA 125 Immunosensor construction process. Reprinted with permission from [90], published by Elsevier, 2020.

**Figure 11 sensors-23-04106-f011:**
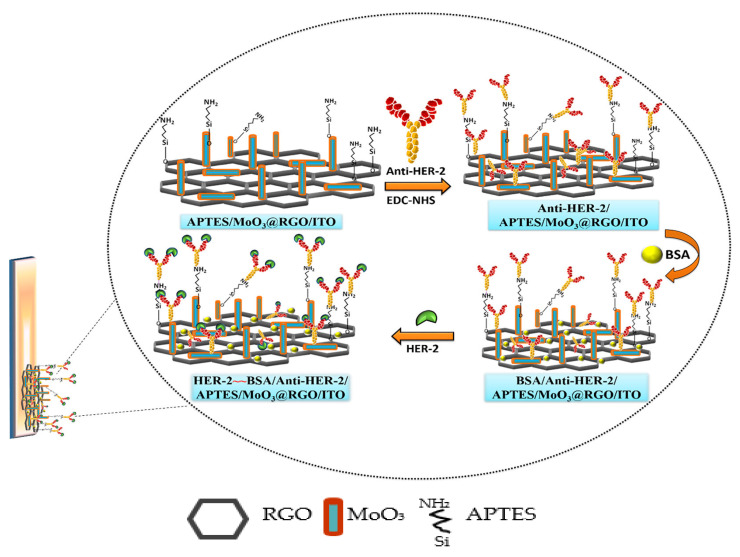
Procedure for developing BSA/anti–HER–2/APTES/MoO_3_@RGO/ITO immunosensor. RGO: reduced graphene oxide; APTES: (3–aminopropyl) triethoxysilane; BSA: Bovine serum albumin; ITO: indium tin oxide electrode; MoO_3_@RGO: graphene oxide doped with molybdenum trioxide. Reprinted with permission from [94], published by American Chemical Society, 2019.

**Figure 12 sensors-23-04106-f012:**
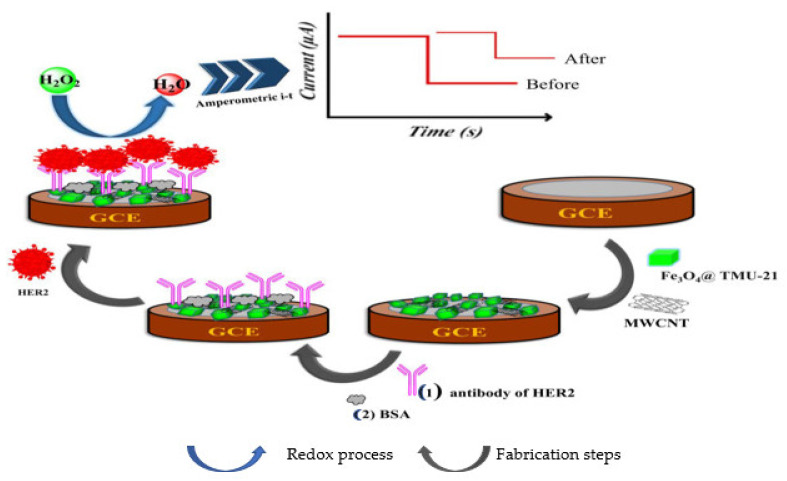
Schematic illustration of the construction process of an immunosensor. Reprinted with permission from [95], published by Elsevier, 2020.

**Figure 13 sensors-23-04106-f013:**
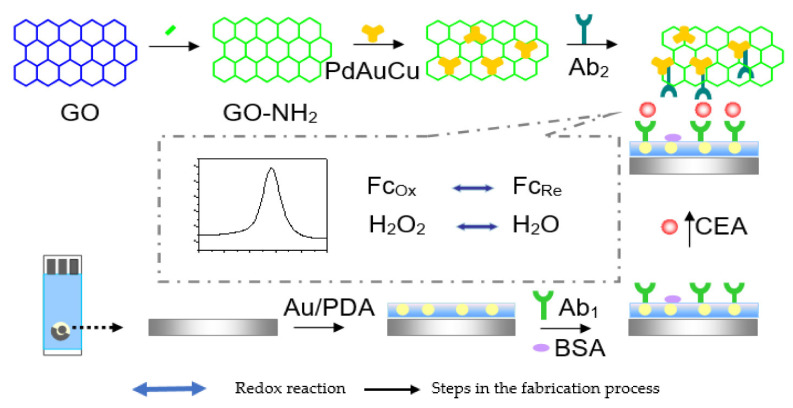
The schematic illustration of the fabrication procedure for CEA immunosensor.GO- NH_2_: amine–doped graphene oxide; Ab1: primary antibody; Ab2: secondary antibody; Fc: feroccene; Ox: oxidation; Re: reduction [108].

**Figure 14 sensors-23-04106-f014:**
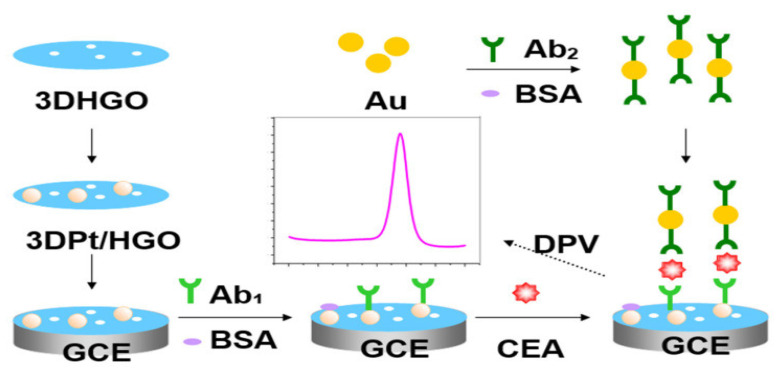
Illustrative fabrication process of carcinoembryonic antigen (CEA) sandwich immunosensor. Ab_1_: capture antibody; Ab_2_: secondary antibody; 3DPt/HGO: 3D porous graphene–oxide–supported platinum metal nanoparticles; GCE: glass carbon electrode [113].

**Figure 15 sensors-23-04106-f015:**
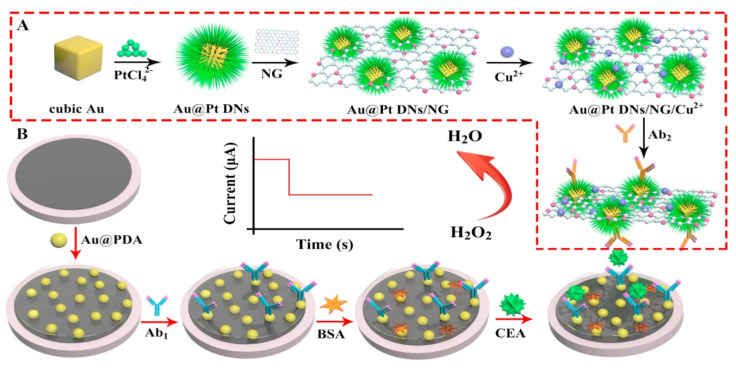
The preparation process of Au@Pt DNs/NG/Cu^2+^–Ab2 and schematic illustration of the sandwich–type EC immunosensor. (**A**): Probe preparation process; (**B**): Electrode modification process for the capturing of antigen and probe. Au@PDA: polydopamine doped Au nanoparticles; Au@PtDNs: cubic Au doped with Pt dendritic nanomaterials; NG: nitrogen–doped graphene. Reprinted with permission from [114], published by Elsevier, 2018.

**Figure 16 sensors-23-04106-f016:**
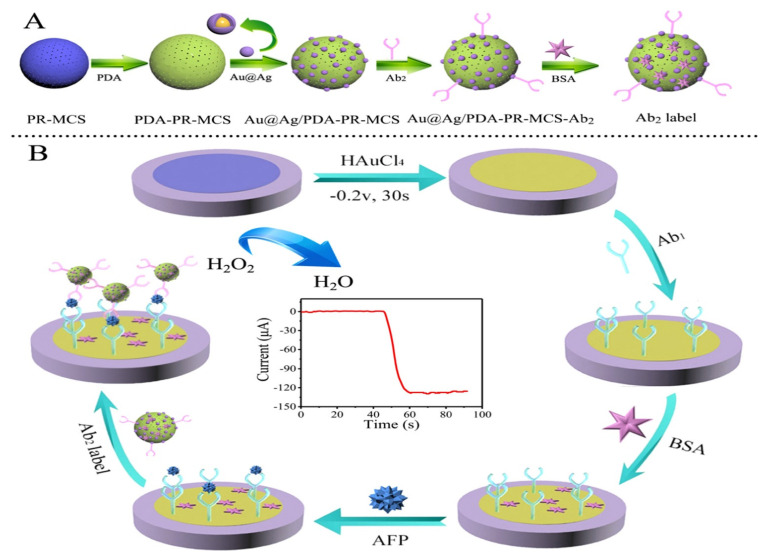
Schematic representation of the fabrication process for Ab2 label and sandwich–type electrochemical immunosensor for AFP detection. (**A**): Probe preparation process; (**B**): Electrode modification process for the capturing of antigen and probe. PDA–PR–MCS: polydopamine–decorated phenolic resin microporous carbon spheres; PR–MCS: phenolic resin microporous carbon spheres. Reprinted with permission from [118], published by Elsevier, 2018.

**Figure 17 sensors-23-04106-f017:**
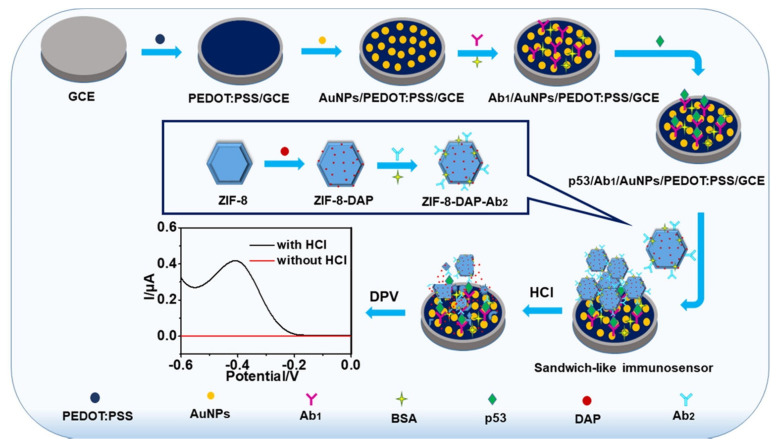
Electrochemical immunosensing process for detecting p53. PEDOT: PSS: poly (3, 4–ethylenedioxythiophene): polystyrenesulfonate; DAP: 2, 3–diaminophenazine; ZIF–8: Zeolitic Imidazolate framework-8; Ab1: primary antibody; Ab2: secondary antibody. Reprinted with permission from [119], published by Elsevier, 2020.

**Figure 18 sensors-23-04106-f018:**
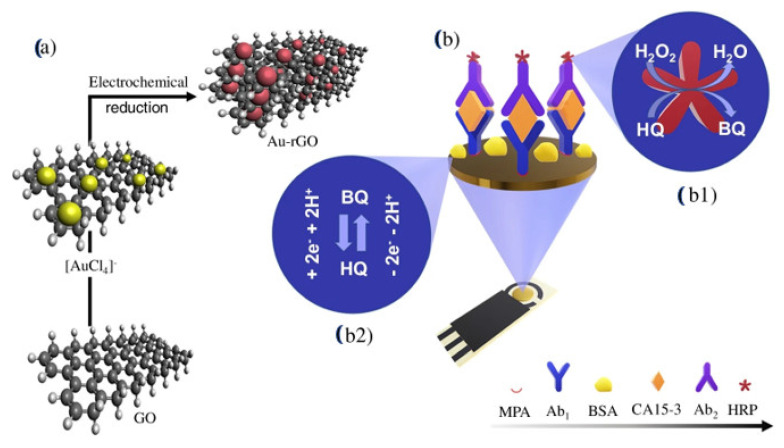
Illustration of the preparation of the electrochemical sandwich–type immunosensor using HRP enzyme for CA15–3 detection. (**a**): preparation process of Au nanoparticles and reduced graphene oxide (Au−rGO) nanocomposite through electrochemical reduction; (**b**): fabrication of the sensor; (**b1**): reduction of H_2_O_2_ by horseradish peroxidase (HRP), using hydroquinone (HQ) as electron mediator: (**b2**): redox process. Reprinted with permission from [120], published by Springer–Verlag GmbH Austria, 2021.

**Figure 19 sensors-23-04106-f019:**
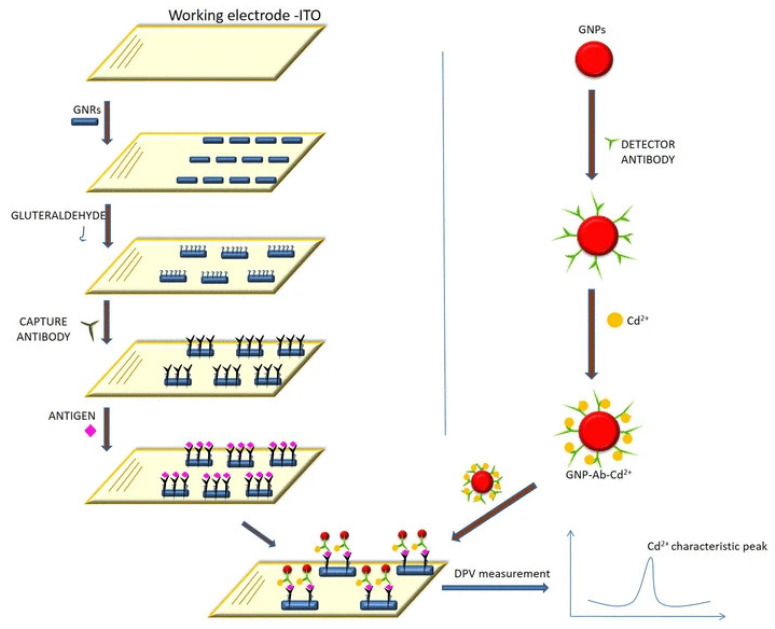
Fabrication scheme for a sandwich–type EC immunosensor and a DPV chart showing the level of attached electrochemical tracer (Cd^2+^) on the AuNPs (GNPs). Reprinted with permission from [122], published by Springer–Verlag GmbH Germany, 2018.

**Figure 20 sensors-23-04106-f020:**
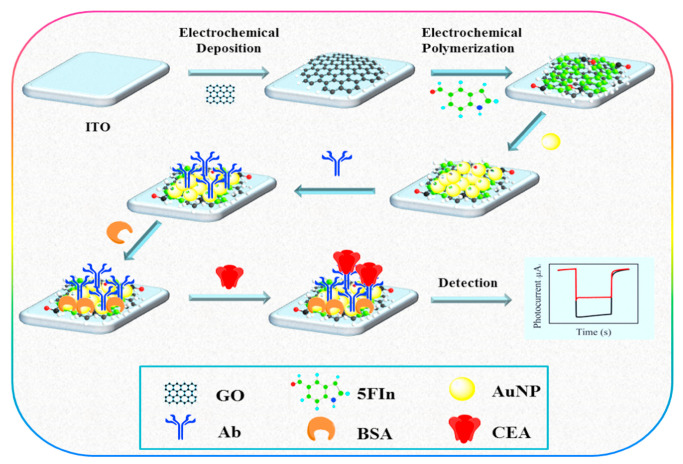
The construction steps in P5FIn/erGO based PEC immunosensor. GO: graphene oxide; Ab: antibody; P5Fln: poly (5–formylindole); 5Fln: 5–formylindole. Reprinted with permission from [135], published by Elsevier, 2018.

**Figure 21 sensors-23-04106-f021:**
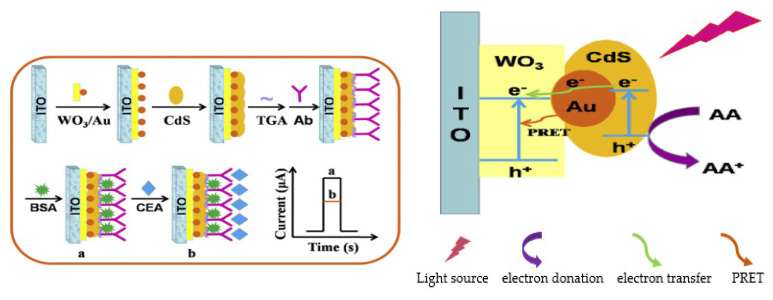
Schematic diagram of the fabrication procedure of a PEC immunosensor and possible charge transfer mechanism at WO_3_/Au/CdS interface in the presence of ascorbic acid (AA) as electron donor. e: electron; h: hole; PRET: Plasmon resonance energy transfer. Reprinted with permission from [136], published by Elsevier, 2018.

**Figure 22 sensors-23-04106-f022:**
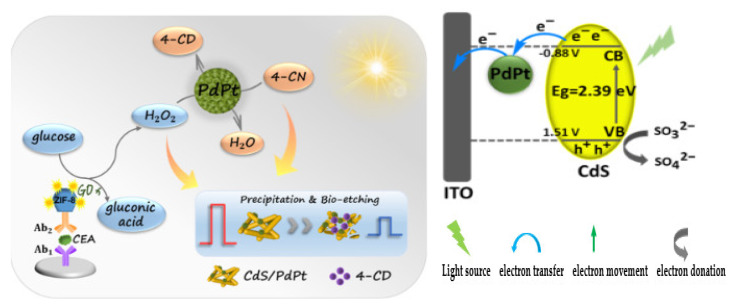
Schematic Illustration for the Detection Mechanism of CdS/PdPt–Based PEC biosensor and band structure of the PEC system under illumination. e: electron; h: hole: VB: valence band; CB: conduction band: Eg: energy gap. Reprinted with permission from [140], published by American Chemical Society, 2021.

**Figure 23 sensors-23-04106-f023:**
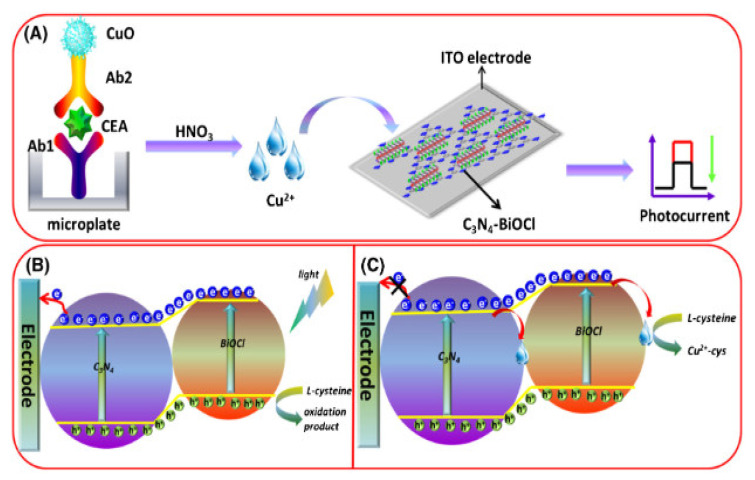
Schematic processes involved in the photoelectrochemical immunoassay of CEA. (**A**): Fabrication process of the biosensor and photocurrent response; (**B**): Photoelectric conversion of C_3_N_4_–BiOCl; (**C**): mechanism of Cu^2+^ -quenched photocurrent of C_3_N_4_–BiOCl in the presence of L-cys. Reprinted with permission from [27], published by Springer–Verlag GmbH Austria, 2019.

**Figure 24 sensors-23-04106-f024:**
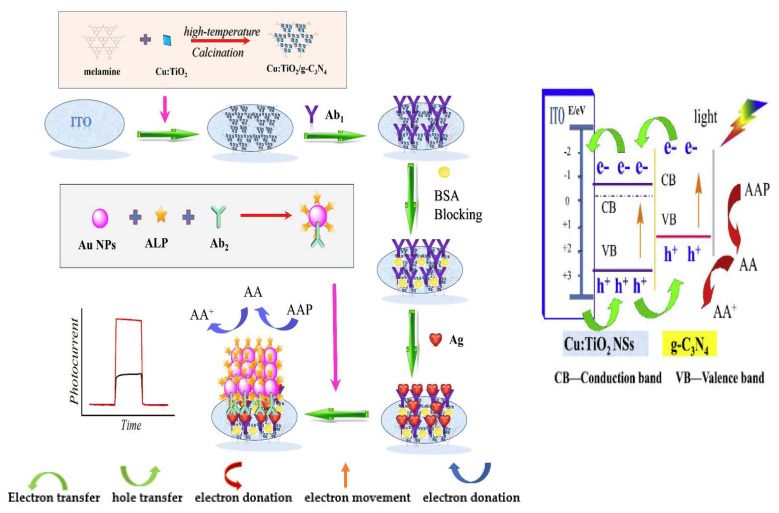
Schematic illustration and electron transfer mechanism of the PEC immunosensor. e: electron; h: hole: AA: ascorbic acid; AAP: ascorbic acid 2–phosphate; Ag: antigen; Ab1: primary antibody; Ab2: Secondary antibody. Reprinted with permission from [28], published by Elsevier, 2019.

**Figure 25 sensors-23-04106-f025:**
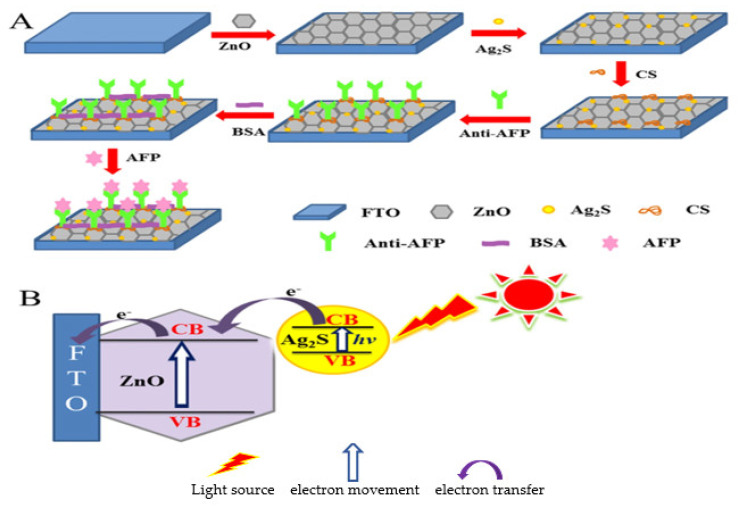
Schematic development process of the immunosensor (**A**) and photocurrent activity of ZnO/Ag2S nanocomposites (**B**). VB: valence band; CB: conduction band; e: electron [149].

**Figure 26 sensors-23-04106-f026:**
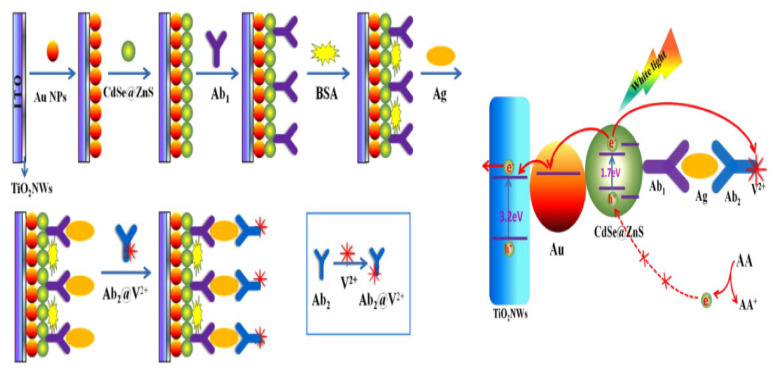
Construction process of the photoelectrochemical immunoassay for CA19–9 detection and photogenerated electron–hole transfer mechanism of the immunosensor for target antigen (Ag) detection. AA: ascorbic acid; e: electron; Ab_1_: primary antibody; Ab_2_: secondary antibody. Reprinted with permission from [154], published by Elsevier, 2015.

**Figure 27 sensors-23-04106-f027:**
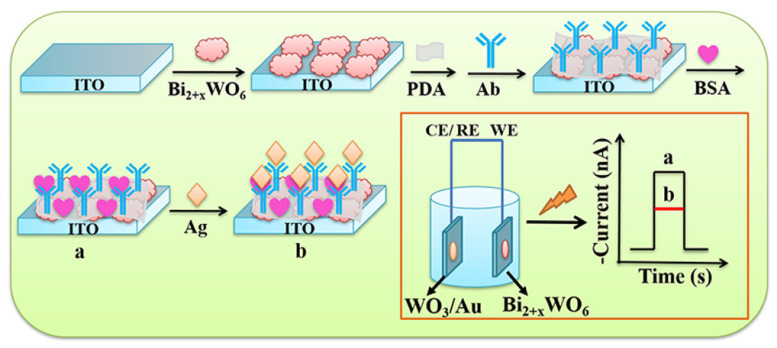
Procedure for the fabrication of Photocathode–Based Immunosensor ITO/Bi_2+x_WO_6_/Ab/BSA/HE4. CE: counter electrode; RE: reference electrode; WE: working electrode; Ab: antibody; Ag: antigen. Reprinted with permission from [156], published by American Chemical Society, 2020.

**Figure 28 sensors-23-04106-f028:**
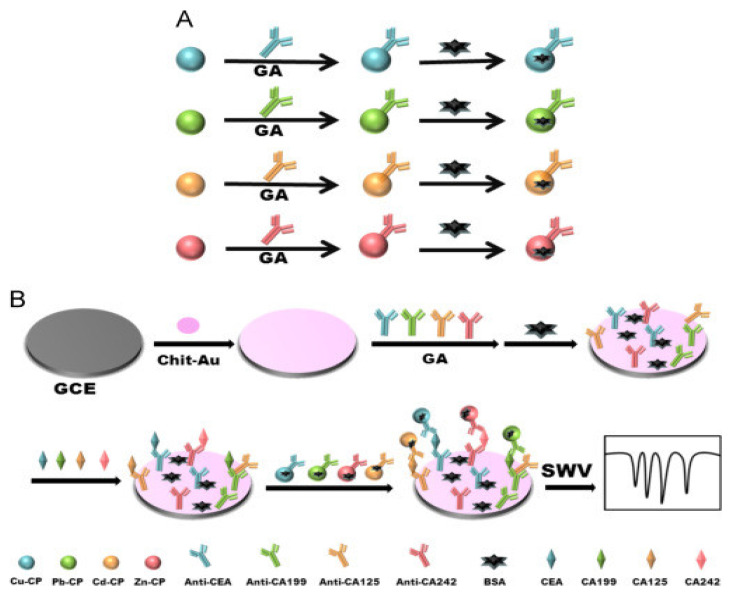
Preparation procedure of Cu–CP–anti–CEA, Pb–CP–anti–CA199, Cd–CP–anti–CA125 and Zn–CP–anti–CA242 immunosensing probes (**A**) and the fabrication steps involved in the sandwich–type immunosensor (**B**). CP: chitosan–poly (acrylic acid) nanospheres; Cu-CP: Cu doped with chitosan–poly (acrylic acid) nanospheres; Pb-CP: Pb doped with chitosan–poly (acrylic acid) nanospheres; Cd-CP: Cd doped with chitosan–poly (acrylic acid) nanospheres; GA: glutaraldehyde; chit-Au: chitosan–decorated AuNPs; SWV: square wave voltammetry. Reprinted with permission from [165], published by Elsevier, 2016.

**Table 1 sensors-23-04106-t001:** Analytical performance of various electrochemical and photoelectrochemical immunosensors in the detection of major ovarian cancer biomarkers.

Biomarkers	Type of Sensor	Electroactive/ Photoactive Materials/Substrate	Label	LOD	Linear Concentration	Ref.
CEA	EC	AuNPs	AgNPs@CS–hemin/rGO–Ab2	6.7 × 10^−6^ ng/mL	2 × 10^−5^ ng/mL to 200 ng/mL	[107]
	EC	Au/PDA	Ab_2_/PdAuCu NPs/Fc–NH_2_–GO	7 × 10^−5^ ng/mL	1 × 10^−4^ to 200 ng/mL	[108]
	EC	T–GO/AuNPs	HRP–(strp–AgNPs)–mAb	7.5 × 10^−5^ ng/mL	1 × 10^−4^ to 5 × 10^−3^ ng/mL	[109]
	EC	rGO/CS/Au NPs	PTh–Au	14.71 × 10^−5^ ng/mL.	0.3 to 30 ng/mL	[110]
	EC	CD–NGs	NiAuPt–NGs	27 × 10^−5^ ng/mL	0.001 to 100 ng/mL	[111]
	EC	GO–AuNPs	Ag@BSA–Pt/Ab2	76 × 10^−5^ ng/mL	0.005 to 100 ng/mL	[112]
	EC	3DPt/HGO	Au–HRP–Ab2	0.0006 ng/mL	0.001–150 ng/mL	[113]
	EC	Au@PDA	Au@PtDNs/NG/Cu^2+^–Ab2	16.7 × 10^−5^ ng/mL	5 × 10^−4^ to 50 ng/mL	[114]
	EC	Au NPs	Au@Ag/Fe_3_O_4_–GS/Ni^2+^	69.7 × 10^−6^ ng/mL	1 × 10^−4^ to 100 ng/mL	[115]
	EC	AuNPs/NB–ERGO	Label–free	0.00045 ng/mL	0.001 to 40 ng/mL	[39]
	EC	Ag/MoS_2_/rGO	Label–free	1.6 × 10^−6^ ng/mL	1 × 10^−5^ to 100 ng/mL	[40]
	EC	Gz–PYSE/PE	Label–free	42.5 × 10^−4^ ng/mL	–	[41]
	EC	AuNPs/CNOs/SWCTNs/CS	Label–free	1 × 10^−4^ ng/mL	1 × 10^−4^ ng/mL to 400 ng/mL	[42]
	EC	NH_2_–G/Thi/AuNPs	Label–free	1 × 10^−2^ ng/mL	5 × 10^−2^ to 500 ng/mL	[37]
	EC	Au@Bi_2_MoO_6_	Label–free	3 × 10^−4^ ng/mL	1 × 10^−3^ to 1 × 10^3^ ng/mL	[43]
	EC	Cu_2_S/Pd/CuO	Label–free	33.1 × 10^−6^ ng/mL	1 × 10^−4^ to 100 ng/mL	[44]
	EC	EG/CNDTs@PPI	Label–free	14.5 × 10^−4^ ng/mL	0.005 to 300 ng/mL	[45]
	EC	MWCNTs/GNPs/HNF	Label–free	0.09 ng/mL	0.4–125 ng/mL	[46]
	EC	mAb/POct pAb/POct	Label–free	10.8 fM 9.08 fM	– –	[38]
	EC	PtPd/N–GQDs@Au	Label–free	2 × 10^−6^ ng/mL	5 × 10^−6^ to 50 ng/mL	[48]
	EC	Au–Ag/rGO@PDA	Label–free	28.6 × 10^−5^ ng/mL	0.001 to 80 ng/mL	[50]
	EC	Au@SiO_2_NPs	Label–free	0.01 ng/mL	0.5 to 10 ng/mL	[47]
	EC	rGO/MoS_2_@PANI	Label–free	3 × 10^−4^ ng/mL	0.001–80 ng/mL	[51]
	PEC	Zn_0_._1_ Cd_0_._9_S/g–C_3_N_4_	Label–free	14 × 10^−4^ ng/mL	5 × 10^−3^ to 20 ng/mL	[133]
	PEC	TiO_2_/CdSeTe@CdS:Mn	Ab2–CuS	16 × 10^−5^ ng/mL	5 × 10^−4^ to 100 ng/mL	[134]
	PEC	P5FLn/erGO	Label–free	14 × 10^−5^ ng/mL	5 × 10^−4^ to 50 ng/mL	[135]
	PEC	WO_3_/Au/CdS	Label–free	1 × 10^−3^ ng/mL	0.01 to 10 ng/mL	[136]
	PEC	Au/WS_2_	Label–free	5 × 10^−4^ ng/mL	1 × 10^−3^ to 40 ng/mL	[137]
	PEC	Au–TiO_2_	CdSe/melamine	5 × 10^−3^ ng/mL	5 × 10^−3^ to 1000 ng/mL	[138]
	PEC	AuNPs	MoS_2_/g–C_3_N_4_–PtCu–Ab2	33 × 10^−6^ ng/mL	1 × 10^−4^ to 80 ng/mL	[139]
	PEC	CdS/PdPt	ZIF–8@GOx–Ab2	21 × 10^−5^ ng/mL	1 × 10^−3^ to 5 ng/mL	[140]
	PEC	C_3_N_4_–BiOCl/PDDA	CuO–Ab2	1 × 10^−4^ ng/mL	1 × 10^−4^ to 10 ng/mL	[27]
	PEC	g–C_3_N_4_/TiO_2_	Label–free	21 × 10^−4^ ng/mL	0.01 to 10 ng/mL	[141]
	PEC	WO_3_@BiOI@CdS	Label–free	32 × 10^−4^ ng/mL	0.01 to 50 ng/mL	[142]
	PEC	Cu:TiO_2_/g–C_3_N_4_	ALP–Au–Ab2	1 × 10^−3^ ng/mL	0.5 × 10^−3^ –1000 ng/mL	[28]
	PEC	2D–ReS_2_	ALP–Ab2	46.8 × 10^−5^ ng/mL	5 × 10^−4^ to 10.0 ng/mL	[143]
	PEC	Au@TiO_2_	CdSe@BiVO_4_–Ab2	5 × 10^−4^ ng/mL	0.01 to 50 ng/mL	[32]
	PEC	g–C_3_N_4_/CdSe –PDDA	Label–free	0.21 ng/mL	10 to 1 × 10^5^ ng/ml	[144]
CA 19–9	EC	CS–MWCNT–Fe_3_O_4_	Label–free	16.3 × 10^−5^ ng/ml	1 × 10^−3^ to 100 ng/mL	[55]
	EC	CeO_2_/FeO_x_@mC	Label–free	1 × 10^−5^ U/mL	1 × 10^−4^ to 10 U/mL	[56]
	EC	GA/CHIT/Zn–Co–S@G	Label–free	0.82 U/mL	6.3 to 300 U/mL	[53]
	EC	Au NPs/Au NPs@PThi	Label–free	0.26 U/mL	6.5 to 520 U/mL	[52]
	EC	CB–PEL	Label–free	0.07 U/mL	0.01 to 40 U/mL	[57]
	EC	CNO/GO	Label–free	0.12 U/mL	0.3–100 U/mL	[58]
	EC	GO–MA	PDA–Ag NPs	3.2 × 10^−5^ U/mL	1 × 10^−4^ to 100 U/mL	[116]
	EC	MPA/ME/Au	Label–free	0.01 U/mL	0.05–500 U/mL	[60]
	EC	PEI/MWCNTs/NHS–EDC	Label–free	0.35 U/mL	–	[59]
	EC	AuNPs@POM	1D MoS_2_ NRs/LNO	0.030 µU/mL	–	[117]
	PEC	TiO_2_ NWs/Au/CdSe@ZnS	Ab2@V^2+^	0.0039 U/mL	0.01 to 200 U/mL	[154]
AFP	EC	PDA–N–MWCNT	NH_2_ –GS/Au@Pt	5 × 10^−5^ ng/mL	1 × 10^−4^ to 10 ng/mL	[14]
	EC	GaN–PDA/Au NPs	Label–free	3 × 10^−3^ ng/mL	0.01 to 100 ng/mL	[66]
	EC	TB–Au–Fe_3_O_4_–rGO	Label–free	27 × 10^−7^ ng/mL	1.0 × 10^−5^ to 10.0 ng/mL	[67]
	EC	Au NPs	Au@Ag/PDA–PR–MCS–Ab2	6.7 × 10^−6^ ng/mL	2 × 10^−5^ to 100 ng/mL	[118]
	EC	Au NPs	Pd/APTES–M–CeO_2_ –GS	33 × 10^−6^ ng/mL	1 × 10^−4^ to 50 ng/mL	[105]
	EC	GO–MB–Au NPs	AuC–HRP–anti–AFP	15 × 10^−4^ ng/mL	5 × 10^−3^ to 20 ng/mL	[65]
	PEC	WS_2_/CdS	HRP–Ab2	43 × 10^−5^ ng/mL	1 × 10^−3^ to 20 ng/mL	[145]
	PEC	TiO_2_ NTs/g–C_3_N_4_	Co_3_O_4_ NPs–Ab2	2 × 10^−4^ ng/mL	4 × 10^−4^ to 40 ng/mL	[146]
	PEC	Au–ZnO	Label–free	56 × 10^−5^ ng/mL	5 × 10^−3^ to 50 ng/mL	[147]
	PEC	CdSe QDs	Bio–anti–AFP/SA/Bio–APOAA	31 × 10^−5^ ng/mL	1 × 10^−3^ to 1 × 10^3^ ng/mL	[148]
	PEC	ZnO/Ag_2_S/CS	Label–free	8 × 10^−3^ ng/mL	0.05 to 200 ng/mL	[149]
	PEC	AuCs_x_WO_3_/CS	Label–free	7 × 10^−3^ ng/mL	0.01 to 500 ng/mL	[150]
	PEC	GO	Ab2@AC60–Gr–GO	54 × 10^−5^ ng/mL	1 × 10^−3^ ng/mL to 100 ng/mL	[151]
p53	EC	P–Cys/GQDs/GNPs–streptavidin/HRP	Label–free	6.5 × 10^−8^ ng/mL	5.92 × 10^−7^ to 12.96 × 10^−4^ ng/mL	[70]
	EC	AuNPs/ERGO	Label–free	88 × 10^−6^ ng/ml	1 × 10^−4^ to 10 ng/mL	[71]
	EC	Chitosan–CB	Label–free	3 × 10^−6^ ng/mL	1 × 10^−5^ to 2 × 10^−3^ ng/mL	[69]
	EC	Star_PGMA_	Label–free	7 × 10^−6^ ng/mL	2 × 10^−5^ to 4 × 10^−3^ ng/mL	[68]
	EC	PEI/NiFe_2_O_4_ NPs	Label–free	5 × 10^−6^ ng/mL	1 × 10^−3^ to 10 ng/mL.	[74]
	EC	NiPc	Label–free	–	1 × 10^−4^ to 0.5 ng/mL	[72]
	EC	AuNPs/PEDOT:PSS	ZIF–8–DAP–Ab2	0.09 ng/mL	1–120 ng/mL	[119]
	EC	PGE/piranha	Label–free	0.01 ng/mL	0.01 to 10 ng/mL	[75]
CA 15–3	EC	Au–rGO	HRP–Ab2	8 × 10^−8^ ng/mL	1 × 10^−7^ to 1 × 10^3^ ng/mL	[120]
	EC	Strp/biotinylated mAb	biotinylated mAb/Strp–MB/biotinylated HRP	15 × 10^−6^ U/mL	50 to 15 × 10^−6^ U/mL	[121]
	EC	CysA/AuNSs/GQDs	Label–free	0.1 U/mL	0.16–125 U/mL	[79]
	EC	CoS_2_–GR–AuNPs	Label–free	0.03 U/mL	0.1–150 U/mL	[80]
	EC	MSA/AuSPEs	Label–free	0.95 U/mL.	1.0 to 1000 U/mL	[77]
	EC	CuS–RGO	Label–free	0.3 U/mL	1.0 to 150 U/mL	[76]
	EC	Au@Pt NCs/Fc–g–CS	Label–free	0.17 U/mL	0.5–200 U/mL	[81]
CA 125	EC	ZnO NRs–Au NPs	Label–free	2.5 ng/µL	–	[84]
	EC	rGO/Thi/AuNPs	Label–free	0.01 U/mL	0.1 to 200 U/mL	[83]
	EC	AuNPs/Cys A/ERGO/PDA	Label–free	0.1 U/mL	0.1 to 400 U/mL	[82]
	EC	AuNPs–PB–PtNP–PANI	Label–free	4.4 × 10^−3^ U/mL	0.01–5000 U/mL	[49]
	EC	PANI/Gr	Label–free	0.923 ng/µL	0.92 × 10^−3^ to 15.2 ng/µL	[86]
	EC	N–rGO@CMWCNTs/CS@AuNPs	Label–free	4 × 10^−5^ ng/mL	1 × 10^−5^ to 100 ng/mL	[88]
	EC	AuNRs	AuNP–Ab–Cd^2+^)	3.4 U/mL	20–100 U/mL	[122]
	EC	3DrGO–MWCNTs–PAMAM/AuNPs	Ab–Suc–Cs@MNPs–TB	6× 10^–6^ U/mL	0.0005–75 U/mL	[123]
	EC	Poly(3–HPA))	Label–free	1.45 U/mL	5 to 80 U/mL	[90]
	EC	CS–AuNPs/MWCNT/GO	AuNP–LOx	0.002 U/mL	0.01 to 100 U/mL	[124]
	EC	GNs/Au/Cysteamine	Label–free	5.5 U/mL	10 to 100 U/mL	[89]
	EC	MZnONF	Label–free	0.00113 U/mL	0.001 to 1000 U/mL	[85]
	PEC	CdTe–CS	Ab2–SiO2@PDA	3×10 ^–7^ U/mL	1× 10^–6^ to 100 U/mL	[152]
	EC	NrGO/CNT and NrGO/CNF	Label–free	–	10 to 32 × 10^−4^ U/mL	[125]
	PEC	CdS/ZNRs/RGO	ABEI–GO@HRP	2.0 × 10^−4^ U/mL	5.0 × 10^−4^ to 500 U/mL	[153]
	EC	AuNPs/RGO	Label–free	4.2 × 10^−5^ U/mL	1 × 10^−4^ to 300 U/mL	[87]
HER2	EC	WO_3_/P–Glu	Label–free	1 × 10^−6^ ng/mL	1 × 10^−6^ to 1 ng/mL	[93]
	EC	APTES/MoO_3_@RGO	Label–free	0.001 ng/mL	0.001–500 ng/mL	[94]
	EC	APTMS–Fe_3_O_4_	Hyd@AuNPs–APTMS–Fe_3_O_4_	2.0 × 10^−5^ ng/mL	5.0 × 10^−4^ to 50.0 ng/mL	[126]
	EC	SPCE/Ab1	Ab2–PbS QDs	0.28 ng/mL	1 to 100 ng/mL	[92]
	EC	MnO_2_ NSs/AuNPs	Au@Ag NRs	16.7 × 10^−6^ ng/mL (DPV) and 33.3 × 10^−6^ ng/mL (chronoamperometry)	50 × 10^−6^ to 100 ng/mL (DPV) and 1 × 10^−4^ to 100 ng/mL (chronoamperometry)	[127]
	EC	Fe_3_O_4_@TMU–24	Pt:CdTe QDs	17.5 × 10^−5^ ng/mL	1 × 10^−3^ ng/mL to 100 ng/mL	[129]
	EC	AuNPs/Cu–MOF	CZTS NPs/Pt/g–C_3_N_4_/anti–HER2–Ab2	3 × 10^−6^ ng/mL	1 × 10^−5^ to 1 × 10^−3^ ng/mL	[128]
	EC	Fe_3_O_4_@TMU–21/MWCNTs	Label–free	3 × 10^−4^ ng/mL.	1 × 10^−3^ to 100 ng/mL	[95]
	EC	AuNPs/MPA	Label–free	2.9 ng/mL.	0 to 10 ng/mL	[91]
	EC	GNPs/APTMS/PEG–NHS–Mal	Label–free	1.20 × 10^−2^ ng/mL	–	[96]
HE4	EC	TiO_2_–rGO/Au@Pd HSs	Label–free	13.33 fM/mL	40 to 60 × 10^6^ fM/mL	[101]
	EC	Fe3O4@SiO_2_@Au MNCs	AgPtCo NDs–Ab2	48.7 × 10^−5^ ng/mL	0.001–50 ng/mL	[131]
	EC	AuNPs/N–doped GNs/FAO	Label–free	1210 ng/mL	1 × 10^4^ to 6.5 × 10^4^ ng/mL	[6]
	EC	AuNRs/NH_2_–GS	Au@PdUSs–Ab2	0.33 pmol/L	1 to 50 × 10^3^ pmol/L	[130]
	EC	Hg–SPCE and Bi–SPCE	CdSe–ZnS QDs–Ab2	12 pM and 89 pM	20 to 40 × 10^3^ pM and 100 to 2 × 10^3^ pM	[132]
	PEC	MWCNTs–PDA–AuNPs	Nb3@nPCN–224	5.6 × 10^−4^ ng/mL	1 × 10^−3^ to10.0 ng/mL	[97]
	PEC	Bi_2+x_WO_6_	Label–free	77.8 × 10^−6^ ng/mL	0.001 to 100 ng/mL	[156]
	PEC	WO_3_/Au/PDA	Label–free	15.6 × 10^−4^ ng/mL	0.01 to 200 ng/mL	[155]
	PEC	AuNPs/CdS NSs	Label–free	10.84 × 10^−4^ ng/mL	0.01 to 200 ng/mL	[159]
MUC 1, CA 153 and HER2	Multiplexed EC	PEI–AuNPs + AQ (MUC 1) TH (CA 153) Ag^+^ (HER2)	Label–free	0.10 to 100 ng/mL (MUC1) 0.10 to 100 U/mL (CA 153) 0.10 to 100 ng/mL (HER2)	0.53 ng/mL (MUC1), 0.21 U/mL (CA 153) and 0.50 ng/mL (HER2)	[162]
CEA, CA153 and CA 125	Multiplexed EC	MB–Chi/GR	Label–free	4 × 10^−5^ ng/mL (CEA) 0.04 mU/mL (CA 153) 0.04 mU/mL (CA 125)	1 × 10^−4^ to 0.1 ng/mL (CEA) 0.10 to 100.00 mU/mL (CA 153) 0.10 to 100.00 mU/mL (CA 125)	[163]
CEA and AFP	Multiplexed EC	Fe_3_O_4_	thionine–4ATP–Au NP (CEA) ferrocene–4ATP–Au NPs (AFP)	0.05–120 ng/mL (CEA), 0.018 ng/mL (AFP)	0.05–100 ng/mL (CEA), 0.012 ng/mL (AFP)	[164]
CEA, CA 199, CA125 and CA 242	Multiplexed EC	CHIT–AuNPs	Cu–CP–anti–CEA; Pb–CP–anti–CA199; Cd–CP–anti–CA125; Zn–CP–anti–CA242	0.02 ng/mL(CEA); 0.4 U/mL (CA 199); 0.3 U/mL (CA 125) and 0.4 U/mL(CA 242)	0.1 to 100 ng/mL (CEA); 1 to 150 U/mL (CA 199); 1 to 150 U/mL (CA 125); 1 to 150 U/mL (CA 242);	[165]

Abbreviations: MWCNTs: multi–walled carbon nanotubes; SWCNTs: single–walled carbon nanotubes CMWCNTs: carboxylated multi–walled carbon nanotubes; Gr: graphene; GO: graphene oxide; MB: magnetic beads/methylene blue; HRP: horseradish peroxidase; ZNRs: zinc nanorods; Ab: antibody; PYSE: 1–Pyrenebutyric–Acid–N–hydroxysuccinimide–ester; GZ: graphene–zirconia; SM: Skim milk; PE: screen–printed carbon electrode; POct: polyoctopamine; mAb: monoclonal antibodies; pAb: polyclonal; NHS: N–hydroxysuccinimide: EDC: 1–ethyl–3–(3 dimethylaminopropyl) carbodiimide; MPA: 3–mercaptopropionic acid; ME: β–mercaptoethanol; APTES: (3–aminopropyl) triethoxysilane; *APTMS*: 3–Aminopropyl(trimethoxysilane); CD–NGs: β–cyclodextrin functionalized reduced graphene oxide nanosheets; 3DHGO: 3–dimentional porous graphene oxide; NG: nitrogen–doped graphene; PtDNs: Pt dendritic nanomaterials; AuNPs@POM: polyoxometalate doped AuNPs; LNO: LiNb_3_O_8_; N–MWCNT: Nitrogen–doped multi–walled carbon nanotube; PDA–PR–MCS: polydopamine–decorated phenolic resin microporous carbon spheres; AuC: gold nanocubes; PEDOT:PSS: poly (3, 4–ethylenedioxythiophene): polystyrenesulfonate; DAP: 2, 3–diaminophenazine; Strp: streptavidin; TB: toluidine blue; NB: Nile blue A; PAMAM: Polyamidoamine; Suc–Cs@MNPs: O–Succinyl–chitosan doped magnetic nanoparticles; LOx: Lactate oxidase; CNF: carbon nanofibers; NrGO: nitrogen–doped reduced graphene oxide; CZTS NP: Cu_2_ZnSnS_4_ nanoparticle; MAA: mercaptoacetic acid; Au@Pd USs: Au–doped Pd urchin shaped nanostructures; NDs: nanodendrites; P5FLn: poly (5–formylindole; ZIF: Zeolitic imidazolate frameworks; GOx: glucose oxidase; *PDDA:* Poly(diallyldimethylammonium chloride); FRs: flower–rods; SA: streptavidin; Bio–APOAA: biotin–capped apoferritin encapsulated ascorbic acid; AC60: fullerenes; PWE: paper working electrode; ABEI: N–aminobuthyl–N–ethylisoluminol; NWs: nanowires; V^2+^: N–(2– carboxymethyl)–N′–methyl–4,4′–bipyridinium; nPCN–224: porphyrin metal–organic framework nanosphere; Nb: nanobody; M–P–A: MWCNTs–PDA–AuNPs; PEI: polyethyleneimine; Chi: chitosan; 4ATP: 4–amiothiophenol; Gr: Graphene; AQ: anthraquinone–2–carboxylic acid; TH/Thi: thionine chloride; CP: chitosan–poly (acrylic acid) nanospheres; CS: chitosan/carbon spheres; CHIT: chitosan; EC: electrochemical; PEC: photoelectrochemical; GNP: gold nanoparticles; CPE: carbon paste electrode; ERGO: electrochemically–reduced graphene oxide; rGO: reduced graphene oxide; CNO: carbon nano onions; SPWE: screen–printed working electrode; GCE: glassy carbon electrode; PE: screen–printed carbon electrode; NH_2_–G: amino–functionalized graphene; NPs: nanoparticles; CNDTs: carbon nanodots; PPI: polypropylene imine dendrimer; EG: exfoliated graphite electrode; HNF: core shell honey fibres; SPGE: screen–printed gold electrode; HRP: horse radish peroxidase; GQDs: graphene quantum dots; GO: graphene oxide; PDA: polydopamine; PANI: polyaniline; mC: mesoporous carbon matrix; GA: glutaraldehyde; G: graphene; PThi: polythionine; CB: carbon black; PEL: polyelectrolyte; SPIDES: screen–printed interdigitated electrodes; CNTs: carbon nanotubes; P–Cys: poly L–cysteine; BSA: bovine serum albumin; Star_PGMA_: star shaped poly(glycidyl methacrylate); NiPc: nickel phthalocyanine; PGE: pencil graphite electrode; mAB: monoclonal antibody; CysA: cysteamine; AuSPEs: gold screen–printed electrode; NCs: nanocrystals; Fc–g–CS: ferrocene grafted chitosan; NRs: nanorods; Gr: graphene; GNs: gold nanostructures; MZnONF: MWCNT–ZnO nanofiber; P–Glu: poly glutamic acid; MSA: mercaptosuccinic acid; MNCs: magnetic nanocomposites; NSs: nanospheres/nanospears; MOF: metal organic framework; Ab2: secondary antibody; GS: graphene sheets; GO–MA: graphene oxide– marked melamine; PB: Prussian blue; Poly(3–HPA): poly (3–hydroxyphenyl acetic acid); PEG: polyethylene glycol; Mal: maleimide; LNO: LiNb_3_O_8_; FAO: fructosyl amino–acid oxidase; CP: chitosan–poly (acrylic acid) nanospheres; HSs: heterostructure; T–GO: thiolated graphene oxide; AuCs_x_WO_3_: gold– cesium tungsten bronze; AuNSs: gold nanospears; Hyd@AuNPs: hydrazine functionalized gold nanoparticles; (Au@Pd HSs): palladium doped gold holothurian–shaped NPs; BiOCl: bismuth oxochloride.

## Data Availability

Not applicable.

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
