# Peer review of "Electrochemical and Photoelectrochemical Immunosensors for the Detection of Ovarian Cancer Biomarkers"

_sensors, 2023, doi:10.3390/s23084106_

Round 1

Reviewer 1 Report

The present review deals with the study of electrochemical and Photoelectrochemical Immunosensors for the Detection Ovarian Cancer Biomarkers. The review is very interesting and it can be accepted after minor revision. 

Comments,

1) Recent works should be considered for the discussion, ref.ACS Biomaterials Science & Engineering 8 (7), 2726-2746, 2022.

2) The quality of the figures should be impoved.

3) There are some typo and English errors and it should be rectified. 

Author Response

Reviewer 1

The present review deals with the study of electrochemical and Photoelectrochemical Immunosensors for the Detection Ovarian Cancer Biomarkers. The review is very interesting and it can be accepted after minor revision. 

Comments,

1) Recent works should be considered for the discussion, ref.ACS Biomaterials Science & Engineering 8 (7), 2726-2746, 2022.

2) The quality of the figures should be improved.

3) There are some typo and English errors and it should be rectified. 

We appreciate the reviewer’s comments. The corrections as stated have been effected as follows;

Firstly, current work that have been referred to and others have been checked and used in the discussion.

Secondly, the quality of the figures has been improved.

Thirdly, the typo and English errors have been checked and corrected. Thank you.

Reviewer 2

The authors summarized recent advances in EC and PEC biosensors for ovarian cancer diagnosis. There are some improvements to be made before publication.

  1. There are some good reviews on EC and PEC biosensing, the authors should cite and compare with this topic in the last paragraph of Introduction.
  2. In Figure 1, the word “Display” was hidden.
  3. Copyright for Figures should be obtained and added.
  4. More discussions should be added for Perspectives.

We thank the reviewer for the observations made. The reviewer’s comments have been dealt with accordingly.

Good reviews on EC and PEC have been consulted and used in the introduction as specified.

The word ‘Display’ has been made visible. While more discussions have been done in the work as directed.

Copyrights for figures used have been obtained and added where necessary. Some figures do not need any permission before use as they are under the Creative Commons Attribution License, CC BY which permits unrestricted use, distribution, and reproduction in any medium, provided the original work is properly cited.

Reviewer 3

The authors carried out the review work title of “Electrochemical and Photoelectrochemical Immunosensors for the Detection Ovarian Cancer Biomarkers”. Although being interesting, I find that there are minor shortcomings with the paper that require addressing prior to this being considered for publication in this journal. I have identified the main points for consideration below:

  1. Firstly, the English of the article should be checked from start to finish.
  2. In the abstract part, the year range and the scanned databases should be added.
  3. In the introduction, the advantages of electrochemical methods should be given in detail. Especially its advantages over chromatographic and spectrophotometric techniques should be examined. The following current articles will guide you.
  • Current Analytical Chemistry, Volume 14, Number 1, 2018, pp. 43-48(6)
  • https://doi.org/10.1080/00032719.2010.512684
  • Current Pharmaceutical Analysis, Volume 16, Number 4, 2020, pp. 367-391(25)
  1. The LOD and linear working range given in Table 1 should be the same unit for a clearer

understanding.

These remarks are highly welcomed. The following have been done,

  • The English has been checked.
  • Year range and databases have been added to the abstract.
  • The advantages of electrochemical methods have been elaborated using the relevant works provided by the reviewer and more.
  • The corresponding LOD and linear working range in Table 1 are now in the same unit.

Reviewer 2 Report

The authors summarized recent advances in EC and PEC biosensors for ovarian cancer diagnosis. There are some improvements to be made before publication.

1.      There are some good reviews on EC and PEC biosensing, the authors should cite and compare with this topic in the last paragraph of Introduction.

2.      In Figure 1, the word “Display” was hidden.

3.      Copyright for Figures should be obtained and added.

4.      More discussions should be added for Perspectives.

Author Response

The authors summarized recent advances in EC and PEC biosensors for ovarian cancer diagnosis. There are some improvements to be made before publication.

  1. There are some good reviews on EC and PEC biosensing, the authors should cite and compare with this topic in the last paragraph of Introduction.
  2. In Figure 1, the word “Display” was hidden.
  3. Copyright for Figures should be obtained and added.
  4. More discussions should be added for Perspectives.

We thank the reviewer for the observations made. The reviewer’s comments have been dealt with accordingly.

Good reviews on EC and PEC have been consulted and used in the introduction as specified.

The word ‘Display’ has been made visible. While more discussions have been done in the work as directed.

Copyrights for figures used have been obtained and added where necessary. Some figures do not need any permission before use as they are under the Creative Commons Attribution License, CC BY which permits unrestricted use, distribution, and reproduction in any medium, provided the original work is properly cited.

Reviewer 3 Report

Dear Editor;

The authors carried out the review work title of “Electrochemical and Photoelectrochemical Immunosensors for the Detection Ovarian Cancer Biomarkers”. Although being interesting, I find that there are minor shortcomings with the paper that require addressing prior to this being considered for publication in this journal. I have identified the main points for consideration below:

1.      Firstly, the English of the article should be checked from start to finish.

2.      In the abstract part, the year range and the scanned databases should be added.

3.      In the introduction, the advantages of electrochemical methods should be given in detail. Especially its advantages over chromatographic and spectrophotometric techniques should be examined. The following current articles will guide you.

·        Current Analytical Chemistry, Volume 14, Number 1, 2018, pp. 43-48(6)

·        https://doi.org/10.1080/00032719.2010.512684

·        Current Pharmaceutical Analysis, Volume 16, Number 4, 2020, pp. 367-391(25)

4.      The LOD and linear working range given in Table 1 should be the same unit for a clearer understanding.

Author Response

Reviewer 3

The authors carried out the review work title of “Electrochemical and Photoelectrochemical Immunosensors for the Detection Ovarian Cancer Biomarkers”. Although being interesting, I find that there are minor shortcomings with the paper that require addressing prior to this being considered for publication in this journal. I have identified the main points for consideration below:

  1. Firstly, the English of the article should be checked from start to finish.
  2. In the abstract part, the year range and the scanned databases should be added.
  3. In the introduction, the advantages of electrochemical methods should be given in detail. Especially its advantages over chromatographic and spectrophotometric techniques should be examined. The following current articles will guide you.
  • Current Analytical Chemistry, Volume 14, Number 1, 2018, pp. 43-48(6)
  • https://doi.org/10.1080/00032719.2010.512684
  • Current Pharmaceutical Analysis, Volume 16, Number 4, 2020, pp. 367-391(25)
  1. The LOD and linear working range given in Table 1 should be the same unit for a clearer

understanding.

We thank the reviewer for the comments and suggestions. We have done the following as suggested.

  • The English has been checked.
  • Year range and databases have been added to the abstract.
  • The advantages of electrochemical methods have been elaborated using the relevant works provided by the reviewer and more.
  • The corresponding LOD and linear working range in Table 1 are now in the same unit.